# Photonic terahertz phased array via selective excitation of nonlinear Pancharatnam-Berry elements

Li Niu[1,8], Xi Feng[1,8], Xueqian Zhang [1] ✉, Yongchang Lu[1], Qingwei Wang[1], Quan Xu [1], Xieyu Chen [1], Jiajun Ma[1], Haidi Qiu[1], Wei E. I. Sha [2], Shuang Zhang [3], Andrea Alù [4,5] ✉, Weili Zhang [6] ✉ & Jiaguang Han [1,7] ✉

Phased arrays are crucial in various technologies, such as radar and wireless communications, due to their ability to precisely control and steer electromagnetic waves. This precise control improves signal processing and enhances imaging performance. However, extending phased arrays to the terahertz (THz) frequency range has proven challenging, especially for high-frequency operation, broadband performance, two-dimensional (2D) phase control with large antenna arrays, and flexible phase modulation. Here, we introduce a photonic platform to realize a THz phased array that bypasses the above challenges. Our method employs 2D phase coding with 2-bit across a broad THz frequency range from 0.8 to 1.4 THz. The core of our design is a pixelated nonlinear Pancharatnam-Berry (PB) metasurface driven by a spatially modulated femtosecond laser for selective excitation of the desired PB elements, allowing precise phase and wavefront control of the emitted THz signals. We showcase the effectiveness of our method through four proof-of-concept applications: single beamforming, dual beamforming, imaging, and vortex beam generation. The realized photonic platform provides a promising pathway for developing broadband phased arrays in the THz regime.

A phased array is a system featuring multiple antennas that can dynamically steer the direction of the radiated electromagnetic beam. This is typically accomplished by introducing time variations or phase delays in the signal paths of each antenna, effectively compensating for differences in the signal paths through free space[1,2]. The versatility and efficiency of phased arrays make them a crucial component to a wide range of advanced technological systems and applications ranging from radar, communication, and astronomy[3–6].

Research on phased arrays can be dated back to the initial development of radio frequency (RF) technologies. Over time, advancements in microwave and millimeter-wave technologies have significantly influenced their evolution[7,8]. Recently, emerging communication technologies operating at millimeter-wave and terahertz (THz) frequencies have attracted widespread attention due to superior bandwidth, directivity and resolution compared to microwaves[9]. In order to enhance the performance of wireless communication systems at these higher frequencies, especially THz frequencies, new phased

[1]State Key Laboratory of Precision Measurement Technology and Instruments, Tianjin University, Tianjin, China. [2]Key Laboratory of Micro-nano Electronic Devices and Smart Systems of Zhejiang Province, College of Information Science & Electronic Engineering, Zhejiang University, Hangzhou, China. [3]Department of Electrical & Electronic Engineering, University of Hong Kong, Hong Kong, China. [4]Photonics Initiative, Advanced Science Research Center, City University of New York, New York, NY, USA. [5]Physics Program, Graduate Center, City University of New York, New York, NY, USA. [6]School of Electrical and Computer Engineering, Oklahoma State University, Stillwater, OK, USA. [7]Guangxi Key Laboratory of Optoelectronic Information Processing, School of Optoelectronic Engineering, Guilin University of Electronic Technology, Guilin, China. [8]These authors contributed equally: Li Niu, Xi Feng. ✉e-mail: alearn1988@tju.edu.cn; aalu@gc.cuny.edu; weili.zhang@okstate.edu; jiaghan@tju.edu.cn

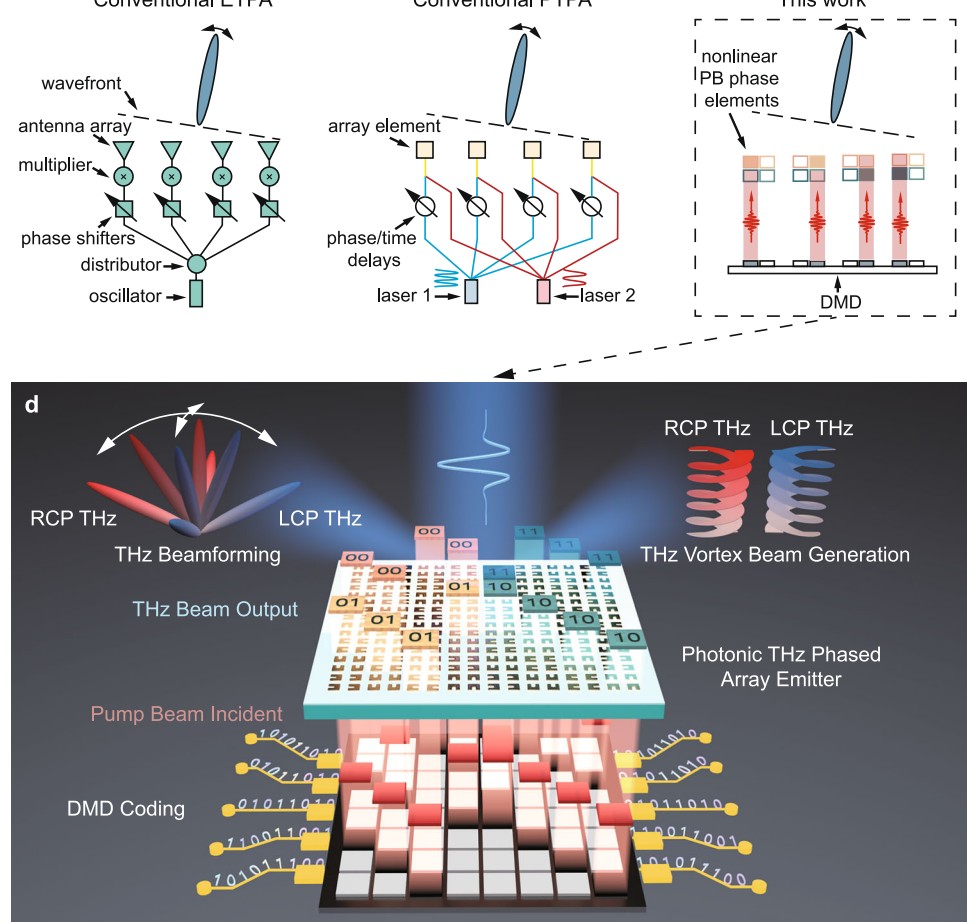

**Fig. 1 | Comparison of different terahertz phased arrays (TPAs) and concept of our photonic TPA (PTPA).** Schematics of conventional electronic TPA (ETPA) (**a**) conventional PTPA (**b**) and this work (**c**). **d** The PTPA here is composed of split-ring resonators (SRRs) with four different orientations, corresponding to 2-bit phase coding to the generated terahertz (THz) waves represented by 00, 01, 10 and 11. By inputting desired 0 and 1 coding patterns to the Digital Micromirror Device (DMD), the angle state of each micromirror can be controlled. The red beam represents the coded near-infrared femtosecond pump beam pattern by the DMD, where only the beam parts reflected by the micromirrors controlled in the schematic black regions can be directed to the PTPA. By matching this beam pattern with SRRs of selected orientations, arbitrary 2-bit phase distribution of the generated THz waves can be generated, thus allowing programmable control over the propagation of the THz waves. The top insets show two proof-of-concept demonstrations of the PTPA, including beamforming and vortex beam generation.

array architectures with superior capabilities are anticipated to play a critical role.

Current THz phased arrays (TPA) mainly include electronic TPAs (ETPAs) and photonic TPAs (PTPAs). Conventional ETPAs operate based on RF electronics and are typically implemented using oscillators, phase shifters, and frequency multipliers[10], as shown in Fig. 1a. Typically, a low-frequency carrier signal (e.g., 90–105 GHz) is generated and distributed to each array element via transmission lines and distributors. Each element contains a phase shifter that adjusts the signal phase, followed by a frequency multiplier (e.g., ×4) to produce a higher frequency output (e.g., 360–420 GHz). This configuration enables high THz power, dynamic and flexible beam steering, and has been widely adopted. However, it is still facing challenges in operating at higher frequencies (e.g., > 1.0 THz), suffering from high insertion loss, electromagnetic interference, phase inaccuracies during signal routing and electronic phase shifting, as well as requiring complex CMOS processing and thermal managements, lacking of phase shifters working for THz waves, etc[11,12].

PTPAs offer a fundamentally different mechanism to bypass the challenges faced by ETPAs through optical frequency down-conversion process[13]. Figure 1b illustrates a typical configuration of PTPA. Two laser signals with frequencies $\omega_1$ and $\omega_2$ are distributed and simultaneously delivered to the array elements for THz generation

with $\omega_{THz} = \omega_1 - \omega_2$. Typical array elements include uni-traveling-carrier photodiode (UTC-PD)[14], photoconductive antenna[15,16], and photomixers[17], etc. The THz phase in each element is tuned by controlling the true phase shift or time delay between the two laser signals by putting an electro-optic phase modulator or delay line into one signal route. This configuration supports high-frequency, tunable operation and holds potential for integration with optical networks but suffers from increased complexity and cost due to the need for multiple physical delay lines and phase control components, especially in future two-dimensional (2D) arrays.

An alternative strategy involves integrating a THz source with a programmable linear metasurface. This approach is based on linear interaction between THz wave and metasurface, and relying on the use of active phase components such as vanadium oxide[18,19], graphene[20], high-electron-mobility transistors[21,22], liquid crystals[23,24], and micromachines[25–27], which have significantly advanced the development of THz manipulation technologies. However, it also faces some issues, such as complex control systems, complicated power supply arrangements, and unwanted interference. Furthermore, the reliance on resonant interactions often restricts the operational bandwidth and limits the phase resolution. Additionally, separating the THz source from the phase control device also introduces additional insertion losses, reducing the system's overall integration[12].

Recently, nonlinear metasurfaces have emerged as a promising solution for precise control of THz wavefronts by utilizing the non-linear Pancharatnam-Berry (PB) phase[28–30]. They have enabled advanced control of broadband THz waves, such as holography[31], polarization[32], and toroidal beam generation[33]. However, implementing TPA requires additional mechanisms for dynamic phase control.

In this work, we introduce an approach for creating a programmable broadband PTPA utilizing nonlinear metasurfaces. Our method employs a digital micromirror device (DMD) to selectively activate specific PB phase control elements in the nonlinear metasurface in a 2D manner, thereby allowing for the customized generation of THz wavefronts, see Fig. 1c. The working mechanism is fundamentally different with ETPAs, and the phase control scheme also differs from that of existing PTPAs. The phase response happens together with the THz generation process and is arisen from abrupt phase change, eliminating the requirements of external phase control devices and the associated synchronization and calibration process. Our method significantly simplifies both the design and control processes for enabling real-time generation of diverse functionalities, and is also extendable into other photonic THz generation platforms. Through proof-of-concept experiments, we demonstrate broadband THz generation up to 0.6-THz bandwidth in full width at half maximum, as well as precise single and dual beamforming, imaging capabilities, and vortex beams creation. These results highlight the potential of this approach for a wide range of THz applications.

## Results

Our PTPA design strategy is illustrated in Fig. 1d. It involves a 2D pixelated structure composed of split-ring resonator (SRR) arrays, with orientations of 0°, 90°, 180°, and 270°. According to nonlinear PB phase[29], for an SRR with an orientation angle $\theta$, regardless of the handedness of the circularly polarized (CP) pump beam, the generated THz wave can be decomposed into left-handed CP (LCP) component with a phase $\theta$ and right-handed CP (RCP) component with a phase $-\theta$ relative to the case of $\theta = 0°$, respectively. This relationship is universal and holds for every difference frequency generation process within the pump spectrum, rendering the nonlinear PB phase naturally broadband for the emitted THz wave (see Supplementary Note 1). Therefore, when pumped by a given CP infrared beam, the generated LCP THz waves from the above SRR arrays acquire phases of 0°, 90°, 180°, and 270°, while RCP THz waves acquire phases of 0°, −90°, −180°, and −270°. This arrangement facilitates a 2-bit PTPA, offering four phase codes: 00, 01, 10, and 11. The utilization of a DMD as a high-speed spatial light modulator enables flexible encoding of the pattern of the pump beam—a near-infrared, circularly polarized femtosecond laser beam—to selectively excite the desired SRR elements. Consequently, advanced programmable broadband THz wavefront control is achieved.

Figure 2a illustrates the basic building block of the PTPA, which consists of a gold SRR fabricated on an indium-tin-oxide (ITO)-coated glass substrate. The geometric parameters are $L_1 = 212$ nm, $L_2 = 220$ nm, $L_3 = 110$ nm, $W = 79$ nm, $t_{SRR} = 40$ nm, $t_{ITO} = 8$ nm, $t_{glass} = 0.7$ mm, and $P_1 = 382$ nm, respectively. This design exhibits a magnetic dipole resonance at 1275 nm (see Supplementary Fig. S2), where the strongest THz emission occurs when the pump wavelength is close to this wavelength due to the resonance field enhancement[29]. The ITO film here is used to boost the THz generation by the broad epsilon-near-zero associated effect[34], and its conductivity can also facilitate the fabrication by mitigating local charges during electron-beam lithography. Each basic PTPA unit is an array of 130 × 130 SRRs, forming a sub-element (with a size of 50 × 50 μm). These sub-elements are then arranged in a 4 × 4 grid, and the SRRs are oriented at 0°, 90°, 180°, and 270°, constituting a PTPA element (with a size of 200 × 200 μm), as depicted in Fig. 2b. The final fabricated PTPA sample, which includes 10 × 5 these elements, covers a total area of

2 × 1 mm (see Methods). A partial scanning electron microscope image of the sample is illustrated in Fig. 2c.

The dynamic 2-bit phase coding mechanism of the PTPA element is depicted in Fig. 2d. It relies on programming a binary coding pattern into the DMD. This pattern modulates the pump beam to selectively excite one of the four sub-elements within the PTPA, leading to the generation of THz waves with a consistent nonlinear PB phase. Although the excited sub-elements are positioned differently within the element, potentially introducing detour phase differences for emitted THz waves towards oblique directions, the main phase contribution is still from the nonlinear PB phase (see Supplementary Note 2). This phase crosstalk is not considered in Fig. 2d for simplicity, which would not affect the fundamental controlling picture.

To better illustrate the DMD coding process, Fig. 2e provides an enlarged view of the micromirror arrangement used to control two sub-elements in the on-state and off-state. Each micromirror has a diagonal length of 10.8 μm. Due to the diamond shape of the micromirrors, the coding area appears square but with slight variations along the edges. During operation, each micromirror alternates between an angle of −12° (on-state) and 12° (off-state), as depicted in Fig. 2f. Only the laser beam that is reflected in the desired direction is collected by the PTPA.

### Dynamic phase control of the PTPA element

To validate the proposed PTPA, we established a DMD-integrated THz time-domain spectroscopy system, as detailed in Supplementary Note 3. This setup enables the projection of various pump beam patterns onto the PTPA, exciting different phase states of the elements and thus creating diverse phase distributions for the generated THz waves. This allows for the direct generation of desired THz wavefronts. Here, all the measurements are performed using an LCP pump. Figure 2g illustrates the measured normalized broadband active phase control response for the generated LCP and RCP THz waves at 1275-nm pump wavelength. In this case, all the elements are sequentially excited using the same patterns shown in Fig. 2d. The phase responses are stably controlled over a broadband frequency range from 0.8 to 1.4 THz. As expected, the phase responses for LCP and RCP THz waves are inverted, with the "00" state serving as the reference. Specifically, the phase of LCP THz waves increases from 0° to 270° in 90° increments, while the phase of RCP THz waves decreases from 0° to −270° in −90° increments. These results confirm the 2-bit broadband phase coding capability of our PTPA element design. The corresponding time-domain THz pulses used to derive these results are shown in Fig. S4.

### Programmable THz single beamforming

We first explore the capability of our PTPA as a THz single beamforming device. The 2D arrangement of the elements in principle allows 2D scanning of the generated THz beam by designing appropriate phase gradients. For simplicity, we focus on a one-dimensional (1D) single beamforming along the $x$-direction as a proof-of-concept demonstration, as illustrated in Fig. 3a. The deflection angle of the beam is controlled by designing an interfacial phase gradient $d\Phi/dx$, which corresponds to the following phase distribution:

$$\varphi(x) = \frac{d\Phi}{dx}x \qquad (1)$$

By discretizing the phase into four levels and the $x$-coordinate according to the resolution of the elements, we can encode the phase using the method in Fig. 2d. Figure 3b, d show two exemplary DMD coding patterns applied to the pump beam, referred as case 1 and case 2, respectively. In case 1, we set $d\Phi_1/dx = 2\pi/\Lambda_1$ with $\Lambda_1 = 1.6$ mm, while in case 2, $d\Phi_2/dx = 2\pi/\Lambda_2$ with $\Lambda_2 = 0.8$ mm. The periodic tilt observed

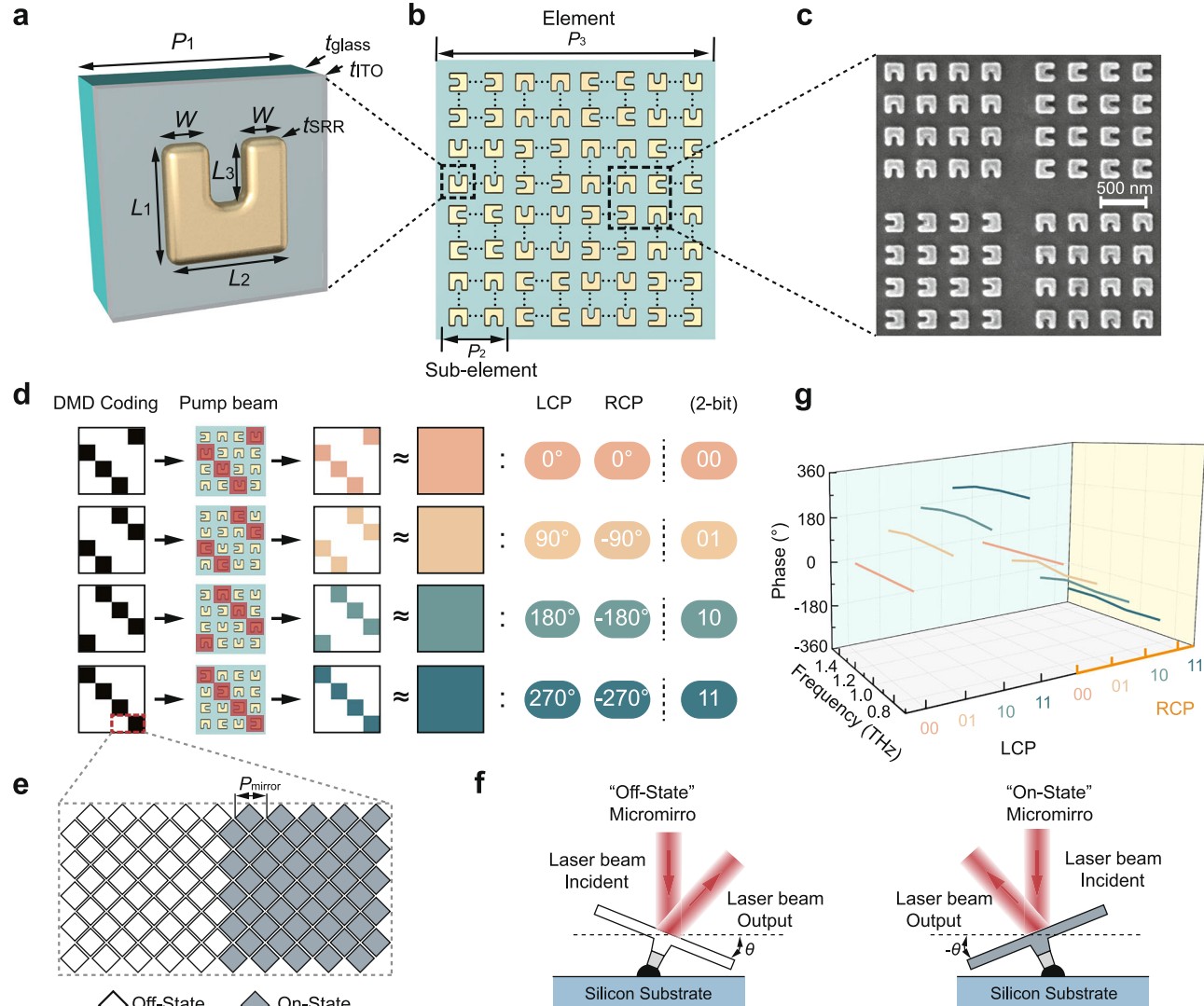

**Fig. 2 | Design of the photonic terahertz phased array (PTPA) element.**
**a** Schematic of the split-ring resonator (SRR) unit cell, which has dimensions of
$L_1 = 212$ nm, $L_2 = 220$ nm, $L_3 = 110$ nm, $W = 79$ nm, $t_{SRR} = 40$ nm, $t_{ITO} = 8$ nm,
$t_{glass} = 0.7$ mm and $P_1 = 382$ nm, respectively. **b** Schematic of the PTPA element,
which is composed of 4 × 4 sub-elements with four different orientations of 0°, 90°,
180° and 270°. Each orientation corresponds to four sub-elements placed in a
staggered arrangement. The periods of each sub-element and element is $P_2 = 50$ μm
and $P_3 = 200$ μm, respectively. **c** Partial scanning electron microscope image of the
fabricated sample. **d** Digital Micromirror Device (DMD) coding patterns to the local

pump beam illuminating on one element, which selectively excited the sub-
elements to achieve 2-bit nonlinear Pancharatnam-Berry (PB) phase control over
the generated left-handed circularly polarized (LCP) and right-handed circularly
polarized (RCP) terahertz (THz) waves. **e** Enlarged view of the micromirrors' states
of the white and black regions in (**d**), corresponding to off-state and on-state,
respectively. **f** The angles of micromirrors working in off-state and on-state.
**g** Measured normalized broadband phase responses of LCP and RCP THz waves
under different DMD coding patterns in (**d**).

in the on-states results from the specific arrangement of sub-elements.
Their corresponding phase coding distributions are depicted in Fig. 3c,
e, which clearly show linear phase variations. For these two cases, the
PTPA exhibits 1.25 and 2.5 phase periods, respectively.

We first carry out numerical calculations to investigate the cor-
responding 1D far-field radiations using[35]:

$$E(\alpha) = \sum_{m=1}^{N_x} \sum_{n=1}^{N_y} I(m,n) \exp\{i[\varphi(m,n) + kP_2(m-1/2)\sin(\alpha)]\} \quad (2)$$

where $N_x$ and $N_y$ denote the number of sub-elements in columns and
rows. $I(m,n) = 0$ or 1 represents whether the sub-element at position
$(m,n)$ is excited, $\varphi(m,n)$ is the phase of the sub-element, $k = 2\pi/\lambda$ is the
wave number with $\lambda$ being the wavelength, and $\alpha$ is the deflection angle
of the generated THz beam.

Figure 3f illustrates the calculated angle-resolved LCP THz
intensity distributions for the two cases at 1.0, 1.2 and 1.4 THz,

respectively. To align with the experimental results, these intensity
distributions have been normalized according to the intensity ratios
of the THz waves generated by the uniform SRR array at different
frequencies (see Supplementary Fig. S2c). The results reveal clear
single THz beamforming. Due to the different magnitudes of the
phase gradients, the deflection angles vary accordingly. Figure 3g
presents the corresponding experimental results, which agree well
with the calculations (see Supplementary Note 4 for experimental
details). The presented range of deflection angle $\alpha$, from –28° to 18°,
is constrained by the numerical aperture of the THz polarizer and
parabolic mirror used for collecting the THz radiation. Both the
calculated and measured deflection angles align with the theoretical
predictions from the generalized Snell's law[36], $\sin(\alpha) = 1/k \times d\Phi/$
$dx = \lambda/\Lambda$, as indicated by the dashed lines in Fig. 3f and g. Taking
1.0 THz as an example, the deflection angles are −10.8° and −22° for
case 1 and case 2, respectively.

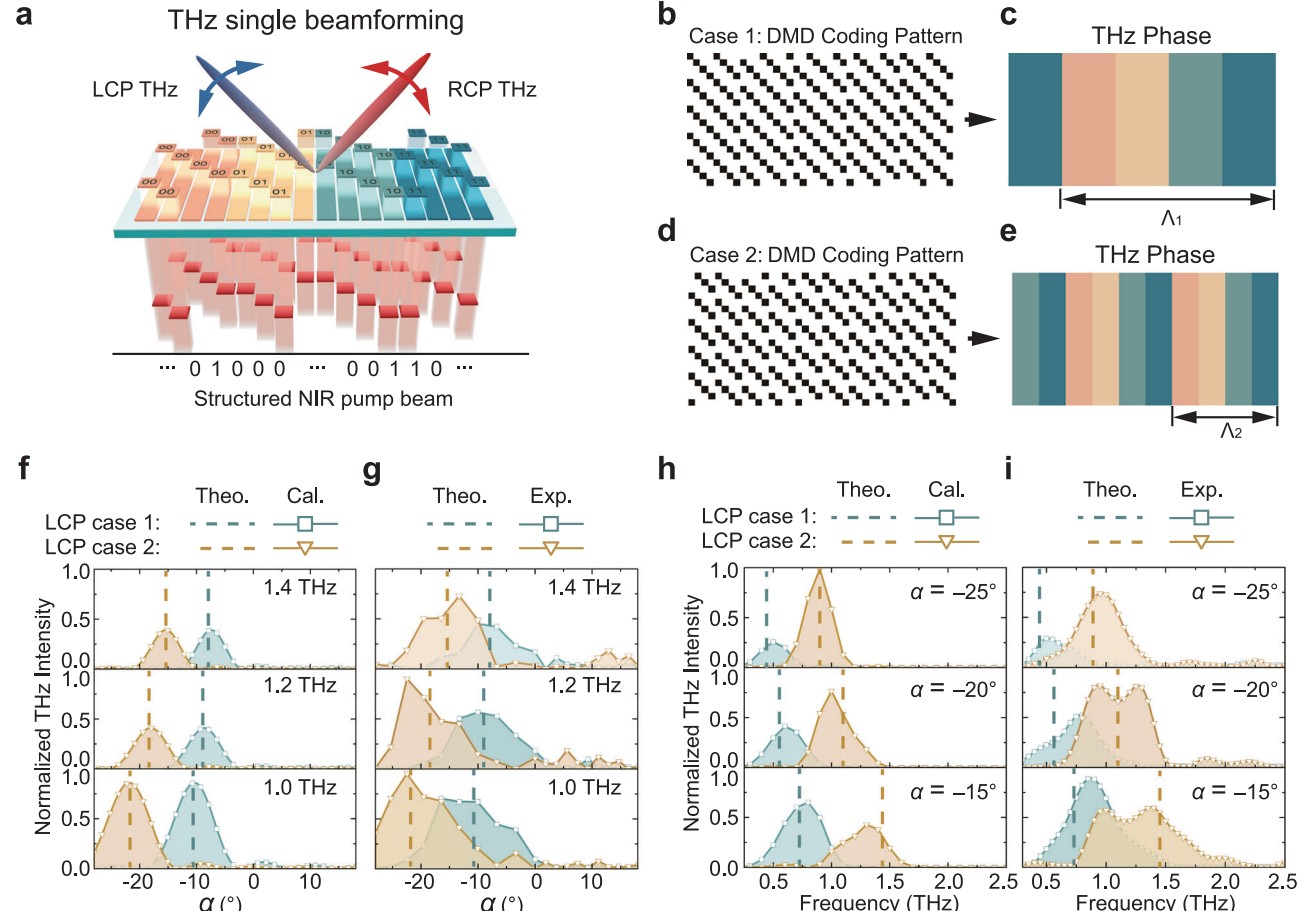

**Fig. 3 | Programmable terahertz (THz) single beamforming. a** Schematic of the photonic THz phased array (PTPA) functionality on single beamforming. **b, d** Digital Micromirror Device (DMD) coding patterns to achieve two different linear phase gradients along the $x$ directions, which are denoted as case 1 and case 2. **c, e** Equivalent phase distributions induced by the DMD coding patterns in **b, d. f, g** Calculated and measured angle-resolved intensity distributions of the generated left-handed circularly polarized (LCP) THz waves at 1.0, 1.2, and 1.4 THz for case 1 and case 2, respectively. **h, i** Calculated and measured intensity spectra of the generated LCP THz waves at −25°, −20°, and −15° for case 1 and case 2, respectively. The dashed lines are theoretical deflection angle and frequency results using generalized Snell's law.

In a phased array, beamwidth $\Omega$ and directivity $D$ are two critical parameters. The beamwidth determines the angular resolution, which reflects the system's ability to distinguish between two closely spaced angular directions. Directivity, on the other hand, reflects the ability of the system to concentrate radiated energy in a specific direction. For our 1D demonstration, the theoretical values are calculated using $\Omega_{\text{theo.}} = \lambda/(L\cos(\alpha))$ and $D_{\text{theo.}} = 4\pi/\Omega_{\text{theo.}}$, where $L$ is the effective aperture length of the PTPA[1]. Larger aperture and smaller deflection angle lead to narrower beamwidth and thereby improved directivity (see Supplementary Note 5). Substituting the relevant values yields theoretical beamwidths $\Omega_{\text{theo.}} = 8.7°$ and $9.3°$, and directivity $D_{\text{theo.}} = 82.3$ and $77.7$ for case 1 and case 2, respectively. In practice, $\Omega$ is defined by the full width at half maximum (FWHM) of the radiated beam. Based on the results shown in Fig. 3f and g, the beamwidth can be estimated as $\Omega_{\text{cal}} = 7.8°$ and $8.3°$, $D_{\text{cal.}} = 92.4$ and $86.6$, while experimental values are $\Omega_{\text{exp.}} = 15.6°$ and $13.8°$, $D_{\text{exp.}} = 46.2$ and $56.1$, respectively. The calculated results are close to the theoretical predictions, implying the effectiveness of our approach in principle. The deviations observed in the experimental results, such as the increased beamwidths and decreased directivities, can be attributed to the limitations of the signal-to-noise ratio and frequency resolution of our current experimental setup. Despite these differences, the values can still serve as useful references without altering the underlying physics. Theoretically, using the coding method is Fig. 2d, the beam steerable range of our PTPA at 1.0 THz can reach [−48.6°, 48.6°]. This can be

expanded by applying non-uniform phase coding method, where the steerable range can reach approximately [−59°, 59°] at 1.0 THz (see Supplementary Note 6). Further enlargement can be achieved using smaller sub-element size $P_2$ to obtain smaller $\Lambda$, which can also refine the angle tuning step.

Additionally, we investigate the frequency bandwidth $\Delta f$ of the two cases at specific deflection angles, which are important for practical applications as they determine the maximum available modulation speed. Figure 3h, i depict the calculated and measured intensity spectra of the generated LCP THz waves at −25°, −20° and −15°, respectively. The main features and trends of these spectra are in good agreement with each other. The dashed lines indicate the theoretical frequencies at corresponding deflection angles. The calculated and experimentally measured frequency bandwidths are $\Delta f_{\text{cal.}}$ ($\Delta f_{\text{exp.}}$) = 0.27 (0.39), 0.35 (0.40) and 0.40 (0.43) THz for case 1, while 0.26 (0.41), 0.31 (0.63) and 0.43 (0.8) THz for case 2, respectively. For the generated RCP THz waves, the above results exhibit deflections towards the opposite angles due to their phase conjugate relation with the LCP THz waves, as shown in Fig. S8.

## Programmable THz dual beamforming

Multiple beamforming enables the generation of multiple THz beams, each directed at different users, facilitating simultaneous services or tasks[37]. This is particularly valuable for enhancing the capacity and efficiency of THz wireless communication and radar applications.

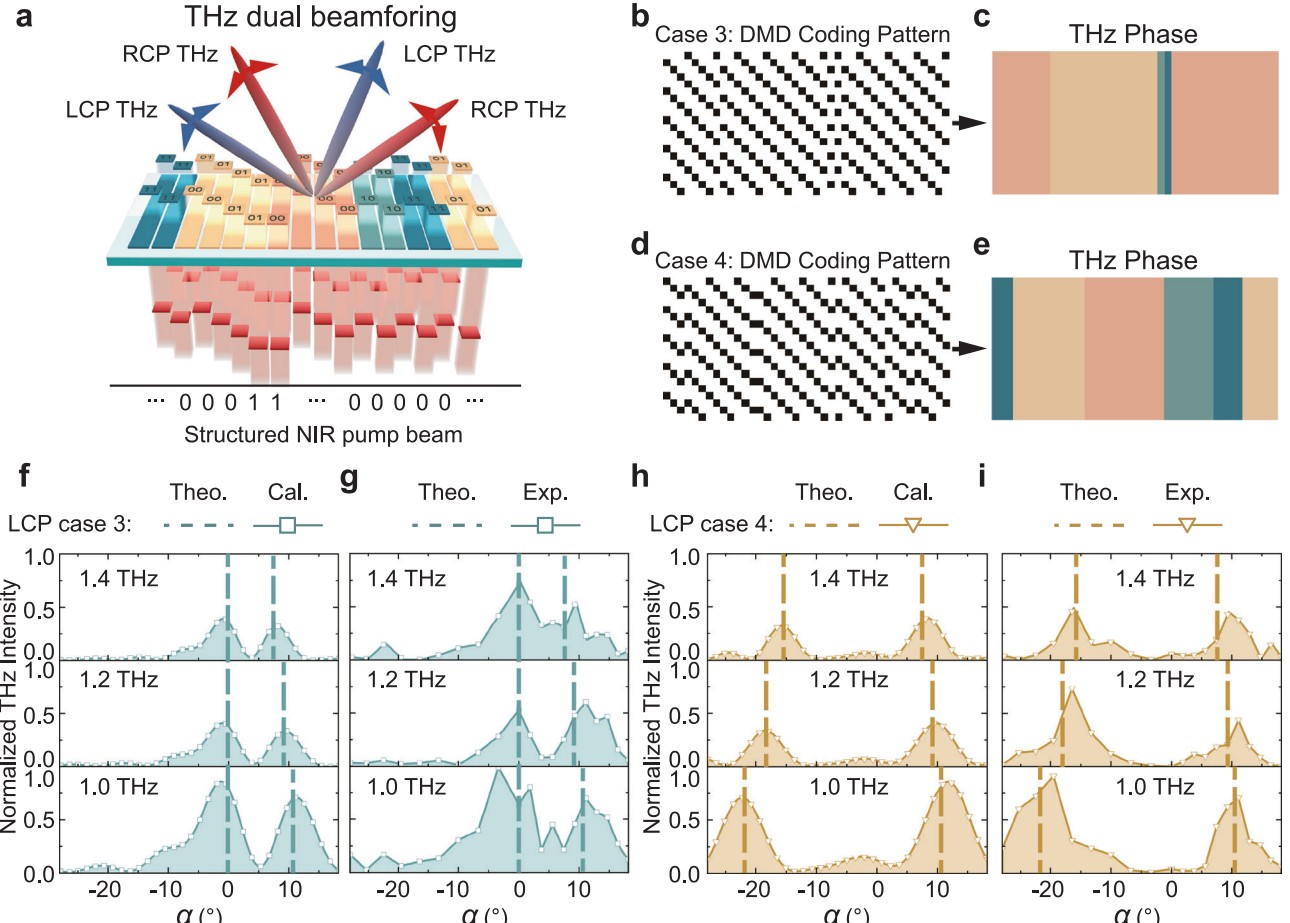

**Fig. 4 | Programmable terahertz (THz) dual beamforming. a** Schematic of the photonic THz phased array (PTPA) functionality on dual beamforming. **b, d** Digital Micromirror Device (DMD) coding patterns to achieve two different phase distributions individually merged by two different phase gradients along the x directions, which are denoted as case 3 and case 4, respectively. **c,e** Equivalent phase distributions induced by the DMD coding patterns in **b**, **d**. **f**, **h** and **g**, **i** Calculated and measured angle-resolved intensity distributions of the generated left-handed circularly polarized (LCP) THz waves for case 3 and case 4, respectively.

Next, we demonstrate the dual beamforming functionality of our PTPA, as shown in Fig. 4a, where we can independently control the deflection angles of the two beams. We use LCP THz beam generation as an example, noting that the RCP counterparts are mirror images of the LCP beams with respect to the normal direction. The crucial aspect lies in establishing appropriate phase distributions within the PTPA, achieved through a holographic method that involves superimposing two obliquely propagating beams as:

$$\varphi(x) = arg\left\{ A_1 \exp\left[i\left(\frac{d\Phi_1}{dx}x\right)\right] + A_2 \exp\left[i\left(\frac{d\Phi_2}{dx}x\right)\right] \right\} \quad (3)$$

where $A_1$, $A_2$ represent the initial amplitudes of the two desired beams, while $d\Phi_1/dx$, $d\Phi_2/dx$ represent their interfacial phase gradients, respectively. Since amplitude control is excluded here, $A_1$ and $A_2$ cannot represent the real amplitude relation between the two output THz beams. To ensure two similar amplitudes, we optimize the far field radiation pattern using Eq. (2), selecting $A_1 = 0.97$ and $A_2 = 1$. Figure 4b, d show two exemplary DMD coding patterns for the pump beam, denoted as case 3 with $d\Phi_1/dx = 0$ mm and $d\Phi_2/dx = -2\pi/1.6$ mm, and case 4 with $d\Phi_1/dx = 2\pi/0.8$ mm and $d\Phi_2/dx = -2\pi/1.6$ mm, respectively. The corresponding equivalent phase coding distributions are illustrated in Fig. 4c, e. The spatially non-uniform on-states and phase distributions result from the interference between the two desired beams.

Figure 4f, h show the calculated angle-resolved LCP THz intensity distributions at 1.0, 1.2, and 1.4 THz, respectively. Clear dual THz beams

with nearly the same amplitude are observed. Figure 4g, i present the measured results. The corresponding deflection angles of them are all consistent with the theoretical calculations, as indicated by the dashed lines. We also compared the theoretical, calculated, and experimental beamwidth and directivity parameters of the generated dual beams at 1.0 THz here. In case 3, we have $\Omega_{\text{theo.1}} = 8.6°$ and $\Omega_{\text{theo.2}} = 8.7°$, $D_{\text{theo.1}} = 83.8$ and $D_{\text{theo.2}} = 82.3$; $\Omega_{\text{cal.1}} = 7.6°$ and $\Omega_{\text{cal.2}} = 6.9°$, $D_{\text{cal.1}} = 94.4$ and $D_{\text{cal.2}} = 104.7$; $\Omega_{\text{exp.1}} = 9.4°$ and $\Omega_{\text{exp.2}} = 7.4°$, $D_{\text{exp.1}} = 76.6$ and $D_{\text{exp.2}} = 97.9$, for the two LCP THz beams deflected to 0° and 10.8°, respectively. In case 4, these parameters are $\Omega_{\text{theo.1}} = 9.3°$ and $\Omega_{\text{theo.2}} = 8.7°$, $D_{\text{theo.1}} = 77.7$ and $D_{\text{theo.2}} = 82.3$; $\Omega_{\text{cal.1}} = 7.9°$ and $\Omega_{\text{cal.2}} = 8.5°$, $D_{\text{cal.1}} = 91.0$ and $D_{\text{cal.2}} = 84.9$; $\Omega_{\text{exp.1}} = 9.2°$ and $\Omega_{\text{exp.2}} = 5.7°$, $D_{\text{exp.1}} = 78.5$ and $D_{\text{exp.2}} = 125.6$, for the two LCP THz beams deflected to –22° and 10.8°, respectively.

## Single-beam steering for slit imaging

Another relevant application of phased arrays is for imaging[38]. THz imaging plays a crucial role in various applications, including active security screening and location detection. Here, we furthermore showcase the capabilities of our PTPA for slit imaging. As shown by the schematic in Fig. 5a, we positioned a metal slit with 5 mm opening after the PTPA (see Supplementary Note 3). Since the slit only permits THz waves to pass through the opening, we can determine its position by gradually adjusting the deflection angle of the generated THz beam. For simplicity, the schematic focuses solely on the beam steering for LCP THz generation. To effectively image the slit, the angle range of the LCP THz beam must encompass the slit area. According to Eq. (1),

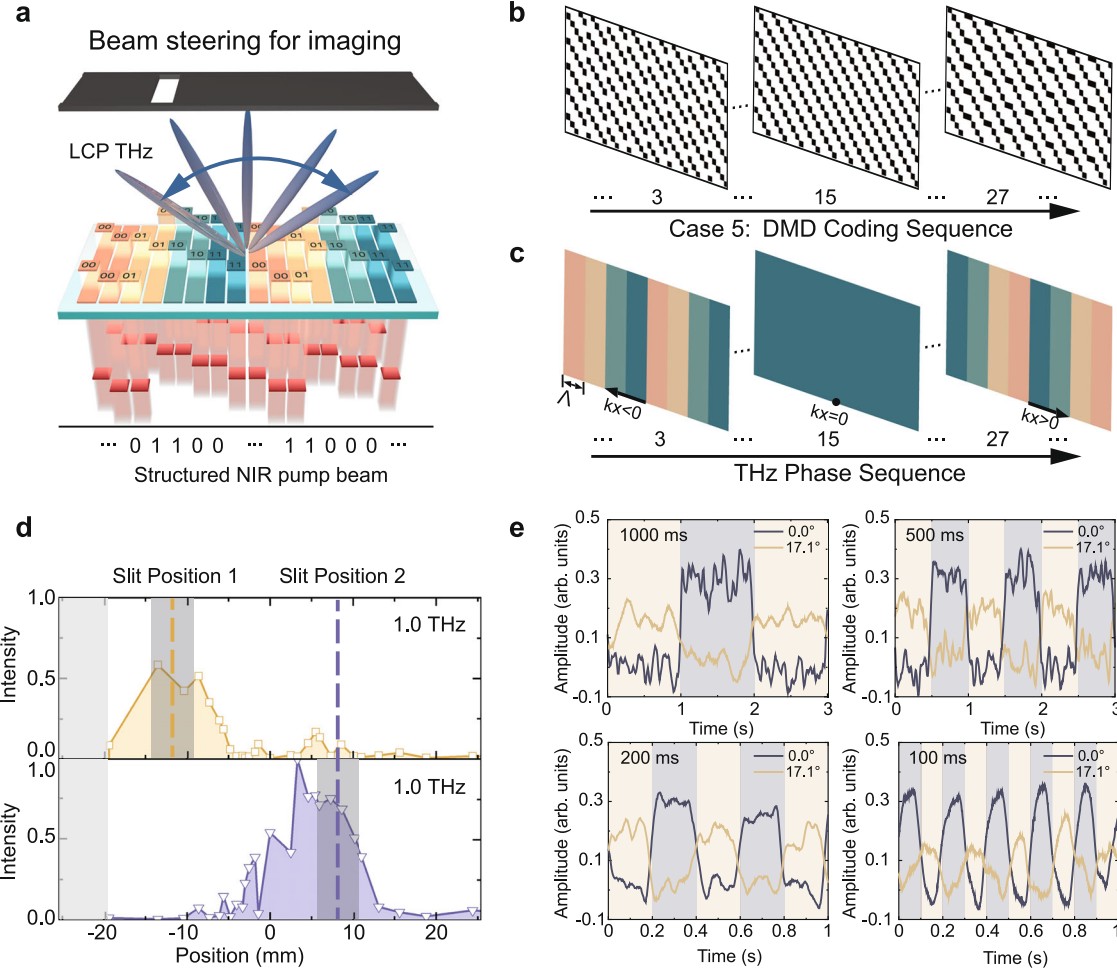

**Fig. 5 | Single beam steering for slit imaging. a** Schematic of the imaging mechanism. **b** Digital Micromirror Device (DMD) coding patterns to achieve gradual terahertz (THz) beam steering along the *x* direction. **c** Equivalent phase distributions induced by the DMD coding patterns in (**b**). **d** Measurements intensity profiles of the generated left-handed circularly polarized (LCP) THz waves at 1.0 THz when the slit are placed at two different positions. The horizontal axis is obtained by transforming the deflection angles at 1.0 THz to positions at the slit plane based on the 29 phase gradients in Tab. S1. **e** Measured variations of the THz time-domain peak amplitudes at 0.0° and 17.1° under different switching periods of the DMD coding patterns (15 and 27), respectively.

we designed 29 single phase gradients to steer the LCP THz beam at 1.0 THz from −29.8° to 29.8°, as detailed in Tab. S1. Figure 5b, c display the 3rd, 15th, and 27th DMD coding patterns and their corresponding phase distributions. In the experiments, these patterns are varied sequentially. Figure 5d shows the measured imaging results of the slit at two different positions. These results are synthesized by extracting the intensity at 1.0 THz and mapping the 29 deflection angles to positions on the slit plane. The gray regions on the left represent areas outside the measurement range of the THz beam with the applied phase gradient. The shaded areas indicate the actual positions of the slit at −12 mm and 8 mm, respectively. The intensity profile clearly reflects the slit positions, although the profiles are wider due to the small size of our PTPA, which causes a broad beamwidth (see Fig. 3f, g) and the diffraction effect of the slit. This setup allows THz signals to be detected beyond the slit area. Increasing the number of elements could improve imaging resolution and reduce such effects.

An important aspect of beam steering is the scanning speed. To assess this metric of performance, we conducted experiments using two DMD coding patterns, indexed by 15 and 27th in Fig. 5b, which support normal and oblique THz emissions. We measured the peak amplitudes with the slit positioned at 0 mm and 18.8 mm, corresponding to the deflection angles of 0.0° and 17.1°, respectively. By holding the delay line at the time-domain peak positions of the THz signals, we observed changes during pattern switching. Figure 5e shows

the relationship between the THz peak amplitudes and the switching periods of 1000 ms, 500 ms, 200 ms, and 100 ms. Clear oscillations are noticeable when the switching period reduces to 100 ms. Ideally, the scanning speed of our PTPA is limited only by the switching speed of the DMD coding patterns. For the applied DMD (DLP4500NIR, Texas Instruments) here, it has a maximum binary pattern rate of 4225 Hz. This can be simply improved using higher-speed DMDs (e.g., DLP650LNIR with 12-kHz refresh rate). However, our experimental setup's performance is constrained by the laser repetition rate (1.0 kHz), chopper speed (370 Hz), and lock-in integration time (30 ms). Consequently, the full potential of our PTPA is not entirely realized. This limitation could be significantly mitigated by using high repetition rate lasers, perfectly continuous wave (CW) lasers for CW THz generation, on the basis of low threshold THz generation and detection mechanisms. Nevertheless, the DMD modulation speed is among an intermediate level of current-related beam steering methods using programmable linear metasurfaces[19,22,23], and is sufficient for many practical applications, such as short-range indoor wireless communication, where beam scanning speed at the kHz level can effectively track users' movement at walking speed. It is also observed that the modulation range measured at 0° is larger because both LCP and RCP THz waves contribute at 0°, whereas only LCP THz waves contribute at 17.1° for pattern 27. Overall, our results effectively demonstrate the potential of our PTPA method for advanced THz imaging applications.

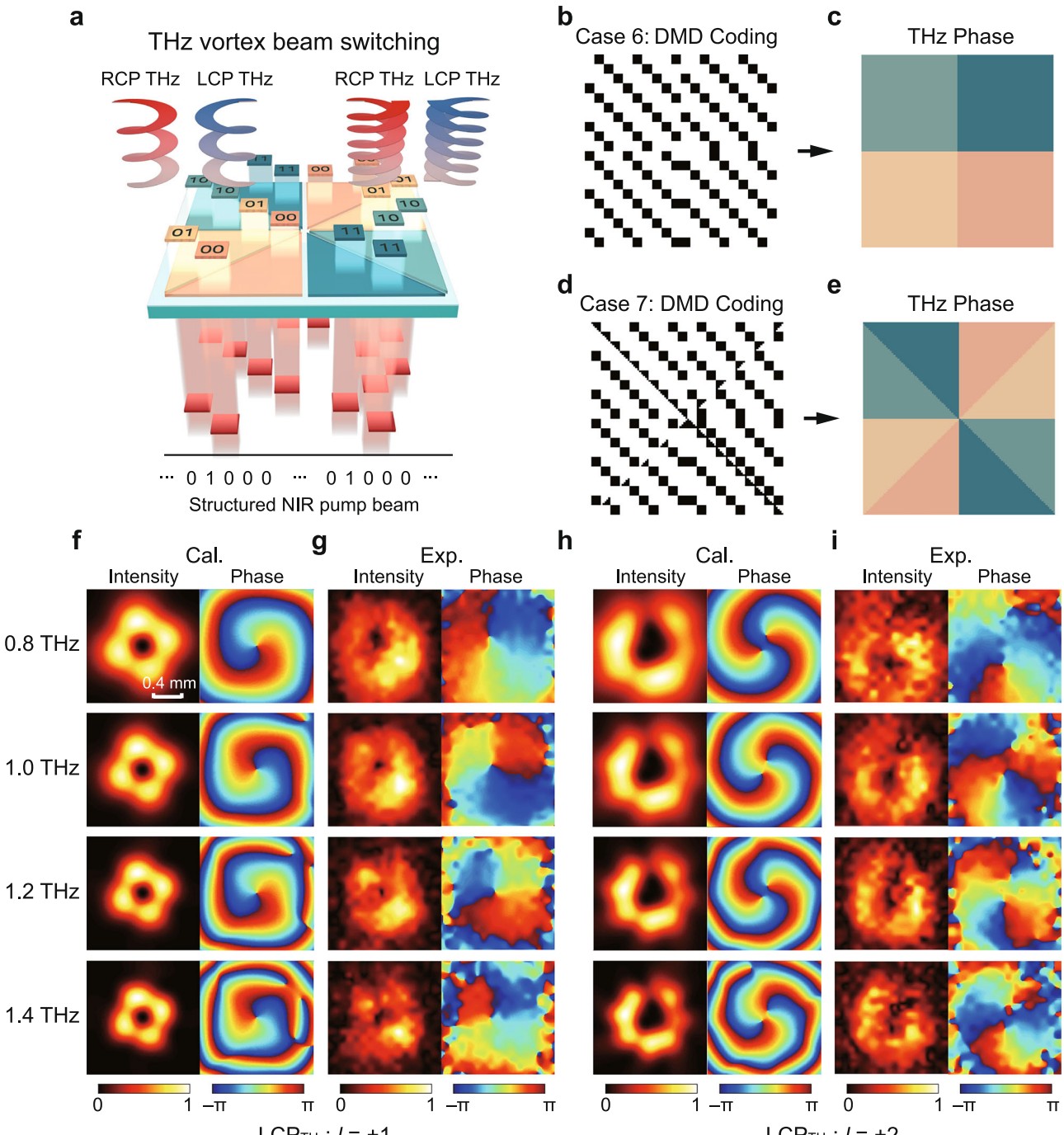

**Fig. 6 | Programmable terahertz (THz) vortex beam generation. a** Schematic of the photonic THz phased array (PTPA) functionality on THz vortex beam generation. **b**, **d** Digital Micromirror Device (DMD) coding patterns to achieve helical phase distributions of topological charges $|l| = 1$ and $|l| = 2$, which are denoted as case 6 and case 7, respectively. **c**, **e** Equivalent phase distributions induced by the DMD coding patterns in **b**, **d**. **f**, **h** and **g**, **i** Calculated and measured transverse intensity and phase distributions of the generated left-handed circularly polarized (LCP) THz beams at 0.8, 1.0, 1.2, and 1.4 THz for case 6 and case 7, respectively. The scalebar in the top-left panel of Fig. 6f is applicable to all the results.

## Programmable THz vortex beam generation

So far, we have demonstrated the 1D phase control capability of our PTPA. To further showcase its 2D phase control ability, we focus on vortex beam generation, which holds promise for enhancing the spatial channels of wireless communications[39]. Our PTPA enables the generation of THz vortex beam carrying controllable topological charge by designing the pattern of the pump beam, as illustrated in Fig. 6a. A vortex beam's momentum consists of spin angular momentum ($s$) and orbital angular momentum ($l$). Here, $s = +1$ and $-1$ correspond to LCP and RCP, while $l$ denotes the topological charge. Since the square area of phase control is sufficient to generate vortex beam, only half of the fabricated PTPA is utilized. The required interfacial phase distribution should follow:

$$\varphi(x, y) = l\phi \quad (4)$$

where $\phi$ represents the azimuth angle of coordinate $(x, y)$. Figure 6b, d illustrates two DMD coding patterns, denoted as case 6 and case 7, for

generating vortex beams with $(s, l)$ = (+1, +1), (−1, −1) and (+1, +2), (−1, −2), respectively. Their equivalent phase distributions are depicted in Fig. 6c, e. These coding patterns effectively divide the PTPA into 4 and 8 azimuthal angle ranges, respectively, with each angle range corresponding to one specific phase code.

Figure 6f, h show the calculated the far-field transverse LCP THz radiation intensity and phase distributions at four exemplary frequencies of 0.8, 1.0, 1.2, and 1.4 THz using Rayleigh-Sommerfeld diffraction theory for case 6 and case 7 at 0.7 mm above the PTPA, respectively[40]. These calculations reveal clear donut-shaped intensity profiles and helical phase loops of $2\pi$ and $4\pi$, corresponding to THz vortex beams of $l$ = 1 and 2. The non-axial symmetric intensity distributions can be attributed to the uneven sampling of the pump beam in different azimuthal angle ranges in the selective excitation, which introduces discrete initial phase distribution. This can be improved using larger working area through increasing the size of the PTPA (see Supplementary Note 7). Figure 6g, i show the experimental results of the generated LCP THz beams, which exhibit the same vortex beam characteristics with the calculations, demonstrating the 2D phase control ability of our PTPA very well. The phase inconsistency observed at the outer edges of the measured vortex beams can be attributed to the corresponding weak intensities, where the influence from the noise naturally enhances. Owing to the small PTPA working area and large divergence angle of vortex beam, a small increase in propagation distance gives rise to a much enlarger beam profile and a phase distribution of more spiral accumulation (see Fig. S10). Nevertheless, this does not affect the consistency of the main feature of the generated vortex beams. Additionally, the measured intensity and phase distributions of the generated RCP THz beams are shown in Fig. S11, illustrating THz vortex beams of $l$ = −1 and −2.

## Discussion

Our results demonstrate the flexibility and programmability of our PTPA up to 0.6-THz bandwidth and 2D operation. Despite limitations in signal-to-noise ratio within our experimental setup, which causes some fluctuations in the measured outcomes, the proof-of-concept demonstrations of our PTPA appear promising for applications. Given that the THz emission cell—or sub-element—is composed of a SRR array, its interaction with adjacent sub-elements is sufficiently minimized to be practically negligible. The nonlinear PB phase underpins the robustness of our phase modulation strategy, allowing us to treat each sub-element as an ideal THz source with a pre-determined phase response. The constraint imposed by the sample finite size forces the generated beams to expand upon emission, in turn limiting the achievable directivity. This constraint can be mitigated through the expedient of increasing the aperture size. To validate the efficacy of our approach, the DMD coding patterns implemented in our experiments are derived through simple and direct phase coding methods. The application of more sophisticated phase coding strategies offers promising avenues for further enhancing the performance of our system.

Limited by both the saturation effect and the low damaging threshold of the proposed nonlinear metasurface, the power conversion efficiency is not high. At a pump fluence of 56.6 μJ/cm$^{-2}$, the efficiency of nonlinear metasurface was estimated to be $6.5 \times 10^{-8}$ by comparing it with a 0.2 mm-thick ZnTe crystal (see Supplementary Note 8). Taking into account the significant thickness difference between the two, this corresponds to an effective second-order non-linear susceptibility ~541 times greater than that of ZnTe. This can be further improved by optimizing the SRR geometry and the ITO film, or trying to transfer the nonlinear PB phase method and the selective excitation concept onto other efficient photonic THz generation platforms with proper designs, such as spintronic emitters[41], photoconductive antennas[42], and photomixers[17], etc. For the purpose of simply increasing the THz power, a straightforward solution is to increase the element number (or overall area) of the nonlinear metasurface, which can scale up the emitted THz power through interference[43]. Both can strengthen the THz radiation, which in turn improves the dynamic range and signal-to-noise ratio of the PTPA, thereby enhancing the beam steering performance.

Additionally, the current nonlinear metasurface platform also faces a challenge in operating under CW optical pump due to the efficiency limitation. However, our working scheme is in principle applicable for CW excitation using other platforms, such as photomixers[44]. As the nonlinear PB phase is a robust phase control method, the radiation phases of LCP and RCP THz waves are solely determined by the orientation angle of the emission structure. This is also true for the photomixer junction with different orientation angles (fixed bias direction). Therefore, by arranging these junctions to form an array with a similar strategy to ours, the proposed PTPA functionalities here can also be realized by employing a similarly selective optical excitation method, or even a selective electric-bias excitation method. Of course, a key challenge of such a device lies in the design of electrode configurations. Recent advances in bias-free two-metal Schottky junctions may provide a pathway to circumvent the need for external voltage sources[45].

The phase control mechanism of our PTPA relies on micro-machine technology. Unlike previous micromachine-based TPAs[25–27], our approach modulates the pump beam but not the external THz wave. Those works rely on the linear interaction between incident THz waves and the devices, following a THz-in/THz-out mode. By contrast, our work follows a pump-in/THz-out mode, which leverages the nonlinear interaction between the optical pump and the device, enabling simultaneous THz generation and manipulation. This interaction naturally supports broadband operation while bypasses the issue of THz insertion and bandwidth loss. Though the size of our PTPA is smaller, which results in larger beam divergence and reduced angular resolution, these can be mitigated through large-area fabrication techniques.

While the current implementation of PTPA uses 2-bit phase coding, the concept can be extended to multiple-bit phase control by increasing the number of sub-elements with more varying orientations within each element. The primary limitations are the resolution and size of the DMD, as well as the THz generation efficiency of the sub-elements. These challenges can be addressed by integrating appropriate imaging systems and introducing efficient mechanisms as mentioned above. In addition to phase coding, simultaneous amplitude coding is achievable by adjusting either the number of excited sub-elements within each element or the number of active micromirrors within each sub-element. The amplitude can be made proportional to the duty cycle of the local pump beam relative to the sub-elements. Alternatively, exciting multiple sub-elements with different orientations can introduce interference and polarization combinations, thereby offering additional degrees of freedom for more versatile THz applications. In terms of future THz communication applications, our PTPA strategy enables the theoretical fusion of optical and THz networks, allowing information transmitted by optical waves to be converted into THz waves. Due to the broadband nature of this method, it supports a significantly larger communication bandwidth and data transfer rate. Moreover, the programmable capability of the PTPA also permits its use as a beamforming device, enabling the delivery of information to one or more targeted end users.

The development of high-frequency phased arrays is of relevant interest. As we push toward THz frequencies, traditional ETPAs and PTPAs both face numerous challenges. In this work, we demonstrated a programmable broadband 2-bit PTPA based on a nonlinear PB phase metasurface. By altering the coding pattern of the pump beam via a DMD, our PTPA achieves real-time 2D wavefront control of THz waves during their generation. Our method supports key functionalities such

as dynamic THz beamforming, precise beam steering for imaging, and vortex beam generation. A significant advantage of our PTPA is its integration of wavefront control directly into the generation process, which eliminates the need for additional THz insertion and mitigates bandwidth losses in the current advanced programmable linear metasurfaces. Furthermore, the principles demonstrated in this study can be applied to other nonlinear THz generation platforms, as well as to linear metasurface platforms controlled by optical pumps. This adaptability enables the realization of similar programmable functionalities across various systems. Our research thus represents a transformative shift in THz technology, providing unprecedented levels of flexibility, adaptability, and functionality for a wide array of THz applications.

## Methods
### Fabrication
The nonlinear metasurface for PTPA was fabricated through a three-step electron beam lithography process, involving electron beam lithography, electron beam evaporation, and lift-off. Initially, the PTPA patterns were created on a 0.7 mm-thick fused quartz substrate coated with an 8 nm-thick ITO layer using a JEOL 9300FS 100 kV EBL tool. The thin ITO layer serves dual purposes of preventing local charge accumulation during lithography and enhancing THz generation. Subsequently, 3 nm-thick Ti and 40 nm-thick Au films were successively deposited via e-beam evaporation. Finally, lift-off and cleaning processes were executed to obtain the PTPA.

## Data availability
The data that support the findings of this study are presented in the paper and the Supplementary Information file.

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

## Acknowledgements

X.Z. was supported by the National Natural Science Foundation of China (Grant No. 62075158). J.H. was supported by the National Natural Science Foundation of China (Grant Nos. 62025504 and 61935015) and the Yunnan Expert Workstation (Grant No. 202205AF150008). Q.X. was supported by the National Natural Science Foundation of China (Grant No. 62135008). X.C. was supported by the National Natural Science Foundation of China (Grant No. 62405215), the China Postdoctoral Science Foundation (Grant No. 2024M752359), and the Postdoctoral Fellowship Program of CPSF (Grant No. GZC20241200). A.A. was supported by the Simons Foundation.

## Author contributions

X.Z. conceived the idea. X.Z., L.N., and X.F. contributed to project conceptualization, methodology, and validation; L.N. and X.F. performed the measurements; L.N. fabricated the samples; L.N. and X.Z. wrote the manuscript with suggestions from X.F., Y. L., Q.W., Q.X., X.C., J.M., H.Q., W.E.I.S., S.Z., A.A., W.Z., and J.H. X. Zhang and J. Han supervised the project. All authors discussed the results and prepared the manuscript.

## Competing interests

The authors declare no competing interests.
