## [Transparent Peer Review file · Nature Communications]

Photonic terahertz phased array via selective excitation of nonlinear Pancharatnam-Berry elements

Corresponding Author: Professor Jianguang Han

Version 0:

Reviewer comments:

Reviewer #1

(Remarks to the Author)

The authors of this manuscript propose a novel approach to realizing a terahertz phased array using photonics instead of electrical control and demonstrate its proof of concept. Their method involves preparing a two-dimensional array of split-ring resonators (SRRs) with controlled orientations, selectively exciting only specific SRRs using a digital micromirror device (DMD), and spatially controlling the phase of the generated terahertz pulses through a nonlinear optical process. This enables control over the propagation of terahertz waves in the far field. To demonstrate this concept, the authors conducted four experiments: single beamforming, dual beamforming, imaging, and vortex beam generation. While the key technologies utilized in this study, such as nonlinear metasurface-based phase-controlled terahertz pulse generation, selective excitation using a DMD, and spatial phase control of the terahertz radiation source for terahertz wave propagation control, are already known, the combination of these techniques in this approach presents a certain degree of novelty. However, several concerns exist, as outlined below. Unless these concerns are adequately addressed, I find it difficult to recommend publication.

1. The significance of this study is unclear. The main novelty of this work lies in replacing the conventional electrical control of active phased arrays with optical control. However, it is not explicitly demonstrated what advantages this provides. For instance, while the authors state that electrical approaches are bulky, the setup shown in Fig. S2 is also quite large. Additionally, the intensity of terahertz waves emitted from nonlinear metamaterials is generally weak, making it unlikely that this method can achieve a practically useful intensity. Moreover, since the modulation speed is ultimately limited by the electrical control speed of the DMD, this approach does not offer a significant speed advantage. Considering these factors, it is difficult to see how this method represents a major breakthrough in the performance of terahertz phased arrays compared to conventional approaches.
2. The explanation of the nonlinear PB phase-based polarization control mechanism for terahertz waves, which is central to this study, is highly insufficient. On page 3, the authors state, "According to nonlinear PB phase, when pumped by a given CP infrared beam, the generated left-handed circularly polarized (LCP) THz waves acquire phases of 0° , 90° , 180° , and 270° , while right-handed circularly polarized (RCP) THz waves acquire phases of 0° , -90° , -180° , and -270° , depending on the orientation of the excited SRR arrays." However, the manuscript does not provide any explanation of why this polarization selection rule arises. A more detailed explanation is necessary so that readers can fully understand the underlying mechanism.
3. The description of the split-ring resonator (SRR) structure used in this study is also insufficient. Is the SRR structure optimal for terahertz beam deflection control using the nonlinear PB phase? Why were these specific structural parameters chosen? Why is an ITO film present? These questions should be clearly addressed.
4. On the last paragraph of page 6, the manuscript states that angle-resolved terahertz intensity was measured, but how was this measurement performed? A clear description of the experimental setup and methodology is needed.
5. Several points regarding the experimental results in Fig. 6 are difficult to understand. Why is the intensity distribution in Fig. 6h not axially symmetric? The manuscript explains that this is due to the discrete nature of the terahertz phase, but even so, the simulation results should still preserve axial symmetry. While it is reasonable that the measured phase distributions in Fig. 6c and Fig. 6g resemble each other, why does the phase distribution in Fig. 6f differ entirely from those in Fig. 6c and

Fig. 6g? The same concern applies to the phase distributions in Fig. 6e, Fig. 6h, and Fig. 6i.

Reviewer #2

(Remarks to the Author)

The authors present a novel technique for terahertz beamforming and steering using a massive array of nonlinear meta-elements with varying orientations, selectively excited by a circularly polarized optical pump. While the reviewer is impressed by the technical advancements, the authors should consider the following points.

1. The title of the paper "Photonic Terahertz Phased Array" appears to be an overstatement and does not accurately reflect the essence of the technical contribution by the authors. Indeed, there have been papers on photonic terahertz phased arrays not cited in the current manuscript such as

-Froberg, Nan Moore, et al. "Terahertz radiation from a photoconducting antenna array." *IEEE Journal of Quantum Electronics* 28.10 (1992): 2291-2301.

-Maki, Ken-ichiro, and Chiko Otani. "Terahertz beam steering and frequency tuning by using the spatial dispersion of ultrafast laser pulses." *Optics Express* 16.14 (2008): 10158-10169.

-Che, Ming, et al. "Arrayed photomixers for THz beam-combining and beam-steering." *Journal of Lightwave Technology* 40.20 (2022): 6657-6665.

I admit that the authors have done a great work to utilize Pancharatnam-Berry (PB) phase, which enables to make the output phase controllable with the orientation of SRRs, while the previous photonic terahertz phased arrays had to rely on the true time delay or the phase shift of the optical pump. The authors should emphasize their technical advancement in comparison to those previous works.

2. The implementation of the present phased array is based on the micromachine technology, i.e. DMD for the light control. In that case, the authors should also discuss the advantages and disadvantages of their approach in comparison with other terahertz phased arrays based on micromachines such as

- Busch, Stefan, et al. "Optically controlled terahertz beam steering and imaging." *Optics letters* 37.8 (2012): 1391-1393.

- Monnai, Yasuaki, et al. "Terahertz beam steering and variable focusing using programmable diffraction gratings." *Optics express* 21.2 (2013): 2347-2354.

- Liu, Xuan, et al. "Terahertz beam steering using a MEMS-based reflectarray configured by a genetic algorithm." *IEEE Access* 10 (2022): 84458-84472.

3. The authors calculate the array factor in Eq. (2) to explain the far-field pattern in specific cases. However, characterizing a phased array requires discussing both the angular resolution and the steerable range of the beam, which must be dependent on the geometry of the SRR array. This analysis is crucial for understanding the necessary improvements to enhance performance in the future.

4. What is the power conversion efficiency of the proposed nonlinear metasurface? The authors should also discuss any potential trade-offs between the beam steering performance and efficiency.

5. The proposed principle may inherently require a short-pulse optical pump, but is there any potential for applying this technique to a CW optical pump? Note that the previous works mentioned above are generally compatible with both pulsed and CW optical pumps.

6. While the authors claim the importance of beam steering in various terahertz applications stating in Line48 "a wide range of advanced technological systems and applications ranging from radar, communication, and astronomy. [3,4]," the cited literature seems to be only about communication, not covering radar and astronomy. The authors might want to add literature on them.

7. The following sentence in Line123 seem to be incomplete.

"Although the excited sub-elements are positioned differently, which induces phase differences for emitted THz waves towards oblique directions."

Version 1:

Reviewer comments:

Reviewer #1

(Remarks to the Author)

The manuscript has been appropriately revised in accordance with my comments.

The novelty of the work as a terahertz light source is now clearly conveyed, and the technical concerns have been resolved. I appreciate the authors' sincere response, and I recommend this manuscript for publication in *Nature Communications*.

Reviewer #2

(Remarks to the Author)

The authors have revised their manuscript, fully addressing the review comments. Hence, I recommend its acceptance.

Response to the Reviewers

We would like to thank all the reviewers for their timely review and the valuable comments, which have helped us improve the manuscript significantly. We have carefully considered each comment and made specific revisions accordingly. Our detailed responses are outlined below.

Reviewer #1:

The authors of this manuscript propose a novel approach to realizing a terahertz phased array using photonics instead of electrical control and demonstrate its proof of concept. Their method involves preparing a two-dimensional array of split-ring resonators (SRRs) with controlled orientations, selectively exciting only specific SRRs using a digital micromirror device (DMD), and spatially controlling the phase of the generated terahertz pulses through a nonlinear optical process. This enables control over the propagation of terahertz waves in the far field. To demonstrate this concept, the authors conducted four experiments: single beamforming, dual beamforming, imaging, and vortex beam generation.

Reply: We sincerely thank the reviewer for the insightful review on our manuscript, particularly for commenting us “propose a novel approach.”

While the key technologies utilized in this study, such as nonlinear metasurface-based phase-controlled terahertz pulse generation, selective excitation using a DMD, and spatial phase control of the terahertz radiation source for terahertz wave propagation control, are already known, the combination of these techniques in this approach presents a certain degree of novelty. However, several concerns exist, as outlined below. Unless these concerns are adequately addressed, I find it difficult to recommend publication.

Reply: We thank the reviewer once again for the positive evaluation of our work, particularly for commenting our approach "presents a certain degree of novelty". We fully acknowledge the reviewer's concerns, and hope that our responses below can adequately address them.

1. The significance of this study is unclear. The main novelty of this work lies in replacing the conventional electrical control of active phased arrays with optical control.

However, it is not explicitly demonstrated what advantages this provides. For instance, while the authors state that electrical approaches are bulky, the setup shown in Fig. S2 is also quite large. Additionally, the intensity of terahertz waves emitted from nonlinear metamaterials is generally weak, making it unlikely that this method can achieve a practically useful intensity. Moreover, since the modulation speed is ultimately limited by the electrical control speed of the DMD, this approach does not offer a significant speed advantage. Considering these factors, it is difficult to see how this method represents a major breakthrough in the performance of terahertz phased arrays compared to conventional approaches.

Reply: We thank the reviewer for pointing this out. As mentioned by the reviewer, we realize a terahertz phased array using photonics instead of electronics. Both types of devices integrate terahertz generation (convert waves in the other frequency ranges to the terahertz frequency range) and terahertz manipulation functionalities together.

Fig. R1. Schematics of conventional ETPA **a**, conventional PTPA **b**, and this work **c**.

1) Significance of our approach

The electronic control here refers to electronic terahertz phased array (ETPA)^{R1}, which operates based on radio-frequency (RF) electronic approaches, as illustrated in Fig. R1a. Typically, a low-frequency carrier signal (e.g., 90 – 105 GHz) is first generated by an oscillator and then distributed to each array element via transmission lines and distributors. Each element is equipped with a phase shifter that adjusts the phase of the incoming signal, which is subsequently passed through a frequency multiplier (e.g., $\times 4$), producing a higher frequency output (e.g., 360 – 420 GHz). This configuration enables high terahertz power, dynamic and flexible beam steering, and has been widely adopted in current terahertz phased arrays (TPAs). *However*, it is still facing challenges in operating at higher frequencies (e.g., > 1.0 THz), suffering from high insertion loss, electromagnetic interference, phase inaccuracies during signal routing and electronic phase shifting, as well as requiring complex CMOS processing and thermal managements, lacking of phase shifters working for terahertz waves, etc^{R2,R3}. For such a configuration, it is hard to transition it from electronic control to optical control owing to the working mechanisms.

In contrast, the optical control here refers to photonic terahertz phased array (PTPA), where the terahertz generation and terahertz manipulation schemes are fundamentally different. The terahertz generation and manipulation are based on frequency down-conversion process (also see the comments of Reviewer #2). This can be intuitively and phenomenally interpreted using the nonlinear polarization equation^{R4}, $P_{\text{THz}}(\omega_{\text{THz}}) = \varepsilon_0 \chi^{(2)} E_1(\omega_1) E_2^*(\omega_2)$, where ε_0 is vacuum permittivity, $\chi^{(2)}$ is second-order nonlinear susceptibility for difference frequency generation (DFG), E_1 and E_2 are two pump laser fields with frequencies ω_1 and ω_2 , ω_{THz} is the generated terahertz frequency satisfying $\omega_{\text{THz}} = \omega_1 - \omega_2$. Figure R1b illustrates a typical configuration of a PTPA. Two laser signals with frequencies ω_1 and ω_2 are distributed and simultaneously delivered to the array elements for terahertz generation through fiber optics. Typical array elements include uni-traveling-carrier photodiode (UTC-PD)^{R5}, photoconductive antenna^{R6}, and photomixers^{R7}, etc. The terahertz phase in each element is tuned by controlling the phase difference between the two laser signals through putting electro-optic phase modulator or delay line into one signal route. Such photonic methods allow high frequency and tunable operations over the generated terahertz waves, as well as hold the potential in integrating into existing optical network. *However*, the duplicate but necessary physical phase modulators and delay lines make the key components bulky and also increase the overall cost, especially for future two-dimensional (2D) beam steering applications.

From some aspects, our PTPA approach indeed replaces the conventional electrical control (electric phase shifters in EPTA, electro-optic phase modulator and delay lines in PTPA) with optical control. *However, the working mechanism is fundamentally different with EPTA, and the phase control scheme is fundamentally different with existing PTPA.* The key of our working scheme lies in **the combination of nonlinear Pancharatnam-Berry (PB) phase and selective excitation concept**, see Fig. R1c, which is experimentally demonstrated using a nonlinear metasurface platform. The phase response happens together with the terahertz generation process and is arisen from abrupt phase change controlled by the orientation of the excited split-ring resonator (SRR, see our response to the 2nd comment of Reviewer #1), eliminating the requirements of external phase control devices and the associated synchronization and calibration process. The selective excitation concept using DMD also enables flexible 2D phased array with extendable phase control level and array size. Meanwhile, the photonic mechanism maintains the broadband and high frequency operation feature.

In short summary, our study does not merely a shift from electrical control to optical control in TPA, but rather presents a new working scheme with distinct and compact phase control method, improved system simplicity, and extendable feature in both design and functionality.

2) Advantages of our approach

We understand the reviewer's concerns on current proof-of-concept demonstration

using nonlinear metasurface platform, which has shorting comings in overall size, terahertz intensity, and modulation speed, etc. ***However, they are not insurmountable in the context of our key working scheme***, which is what we really want to stress.

2.1) Concern on overall size. Yes, if we consider all the building blocks of the overall system containing terahertz generation, propagation, and detection, our setup is indeed big, see Fig. S3. Actually, our initial idea of such a comparison with electrical approach is on the key component of the phased array. In conventional ETPA, the architecture includes phase shifters, frequency multipliers, and extensive microwave circuitry, making the overall device bulky and complex. Likewise, in conventional PTPA, the architecture includes phase modulators/delay lines and terahertz generation elements. By contrast, the key component of our architecture is merely a planar nonlinear metasurface, which is more compact. Although the DMD and the associated optics add to the overall size, the working scheme substantially reduces much hardware requirements for the phase control.

In principle, the selective excitation scheme using DMD can be potentially realized in more compact forms. For example, the pump beam can also be spatially edited to enable selective excitation by integrating liquid crystal (LC) components directly onto the metasurface^{R8} and trying to add local modulation ability of LC^{R9}.

We acknowledge that our previous description may not have made this distinction and further solution clear, and we have now revised the manuscript to emphasize this point more explicitly.

2.2) Concern on terahertz intensity. It is true that the terahertz intensity here is weak owing to the relatively low conversion efficiency (see our response to the 4th comment of Reviewer #2). This is actually a common problem of the current nonlinear metasurface platform and requires further exploration. However, this does not detract from the novelty and feasibility of our approach.

A straightforward solution is to increase the element number (or overall area) of the nonlinear metasurface, which can scale up the emitted power. Theoretically, for nonlinear metasurface composed of N elements, the total THz intensity is proportional to N^2 under ideal coherent combination conditions^{R10}.

More significantly, the proposed working scheme is not limited to nonlinear metasurfaces, which can be transferred to the other THz generation platforms with inherently higher efficiencies. For instance, spintronic emitters have recently demonstrated robust and broadband THz generation^{R11}, photoconductive antennas^{R12} and photomixers⁷ can offer much higher output powers. We believe that the integration of our selective excitation scheme with these platforms upon subtle antenna design could inspire new pathways toward high-performance PTPAs for either pulsed or

continuous-wave (CW) terahertz beam control, such as antenna array with spatially distributed junctions of different orientations.

2.3) Concern on modulation speed. As the reviewer said, the modulation speed of our current implementation is ultimately limited by the DMD. The max refreshing rate of our applied DMD (DLP4500NIR@Texas Instruments) is approximately 4 kHz. Other commercially available DMDs, such as DLP650LNIR, can operate at speed up to 12 kHz. To evaluate the modulation speed level of our PTPA, we can compare it with other terahertz beam steering approaches, such as dynamic linear metasurfaces which only perform terahertz manipulation functionality and require external terahertz sources. Taking LC-, VO₂-, and high electron mobility transistor (HEMT)-based terahertz metasurfaces as examples, the typically reported modulation speeds are on the orders of Hz^{R13}, kHz^{R14}, and GHz^{R15} levels, respectively. Our demonstration is clearly among the mediate level, which is sufficient for certain application scenarios. For example, short-range indoor wireless communication, where beam steering with kHz-level update rates is enough in responding to the movement of a user at walking speed.

In principle, the working scheme is compatible with any technique capable of local optical intensity modulation for selective excitation. Emerging platform includes lithium niobate electro-optic modulators, which can achieve modulation speed of tens of GHz^{R16}. Nevertheless, implementing such solutions at the current stage would increase the overall system complexity.

At last, *while overall size, intensity, and modulation speed are three critical metrics for evaluating TPAs, they are not the sole determinants.* Other essential factors, such as operating frequency, bandwidth, and the ability to achieve 2D control, are also very important metrics in defining the performance of TPAs. The working scheme of our PTPA not only has certain advantages in these aspects, but also offers solutions in overcoming the above weaknesses by combining with other photonic platforms. We believe our work represents a new research direction of PTPA.

We are grateful for the reviewer's thoughtful comments, which have allowed us to further clarify and emphasize our contributions. We have added the above necessary information in the revised manuscript (see **page 2, paragraph 4; page 3, paragraphs 1-3; page 4, paragraph 2; page 12, paragraph 1; page 14, paragraph 3; page 15, paragraph 1; page 19, Fig. 1**).

2. The explanation of the nonlinear PB phase-based polarization control mechanism for terahertz waves, which is central to this study, is highly insufficient. On page 3, the authors state, "According to nonlinear PB phase, when pumped by a given CP infrared

beam, the generated left-handed circularly polarized (LCP) THz waves acquire phases of 0° , 90° , 180° , and 270° , while right-handed circularly polarized (RCP) THz waves acquire phases of 0° , -90° , -180° , and -270° , depending on the orientation of the excited SRR arrays.” However, the manuscript does not provide any explanation of why this polarization selection rule arises. A more detailed explanation is necessary so that readers can fully understand the underlying mechanism.

Reply: We thank the reviewer for pointing this out. Yes, the nonlinear PB phase is the central phase control concept in this study, and we overlook providing sufficient explanation at that place. Inspired by the reviewer’s valuable suggestion, we find that including a more rigorous explanation is very necessary in improving the manuscript’s completeness and readability, instead of just mentioning related references^{R17-R19} once in the introduction.

For a second-order nonlinear DFG process under normal pump, the corresponding effective second-order nonlinear polarization can be expressed as^{R4}:

$$\begin{bmatrix} P_x(\omega_{\text{THz}}) \\ P_y(\omega_{\text{THz}}) \end{bmatrix} = \epsilon_0 \begin{bmatrix} \chi_{xxx}^{(2)} & \chi_{xxy}^{(2)} & \chi_{xyx}^{(2)} & \chi_{xyy}^{(2)} \\ \chi_{yxx}^{(2)} & \chi_{yyx}^{(2)} & \chi_{yyx}^{(2)} & \chi_{yyy}^{(2)} \end{bmatrix} \begin{bmatrix} E_{1x}(\omega_1)E_{2x}^*(\omega_2) \\ E_{1x}(\omega_1)E_{2y}^*(\omega_2) \\ E_{1y}(\omega_1)E_{2x}^*(\omega_2) \\ E_{1y}(\omega_1)E_{2y}^*(\omega_2) \end{bmatrix}. \quad (\text{R1})$$

where the subscripts x, y of nonlinear terahertz polarization P and pump laser field E represent their corresponding polarization components, the subscripts of nonlinear susceptibility ijk represent the generation process of i -polarized nonlinear terahertz polarization from j -polarized pump laser field E_1 and k -polarized pump laser field E_2 with $i, j, k \in \{x, y\}$. Due to the mirror symmetry of the SRR (suppose the mirror plane is along the y direction), the second-order susceptibility tensor can be reduced by $\chi^{(2)}_{xxx} = \chi^{(2)}_{yxy} = \chi^{(2)}_{yyx} = \chi^{(2)}_{xyy} = 0$. Then, Eq. (R1) can be calculated as:

$$\begin{bmatrix} P_x(\omega_{\text{THz}}) \\ P_y(\omega_{\text{THz}}) \end{bmatrix} = \epsilon_0 \begin{bmatrix} 0 & \chi_{xxy}^{(2)} & \chi_{xyx}^{(2)} & 0 \\ \chi_{yxx}^{(2)} & 0 & 0 & \chi_{yyy}^{(2)} \end{bmatrix} \begin{bmatrix} E_{1x}(\omega_1)E_{2x}^*(\omega_2) \\ E_{1x}(\omega_1)E_{2y}^*(\omega_2) \\ E_{1y}(\omega_1)E_{2x}^*(\omega_2) \\ E_{1y}(\omega_1)E_{2y}^*(\omega_2) \end{bmatrix}. \quad (\text{R2})$$

Consider our case that the SRR is rotated by θ , see Fig. R2, Eq. (R2) is still satisfied in its local coordinate $x'y'z'$, i.e.,

$$\begin{bmatrix} P_{x'}(\omega_{\text{THz}}) \\ P_{y'}(\omega_{\text{THz}}) \end{bmatrix} = \epsilon_0 \begin{bmatrix} 0 & \chi_{x'x'y'}^{(2)} & \chi_{x'y'x'}^{(2)} & 0 \\ \chi_{y'x'x'}^{(2)} & 0 & 0 & \chi_{y'y'y'}^{(2)} \end{bmatrix} \begin{bmatrix} E_{1x'}(\omega_1)E_{2x'}^*(\omega_2) \\ E_{1x'}(\omega_1)E_{2y'}^*(\omega_2) \\ E_{1y'}(\omega_1)E_{2x'}^*(\omega_2) \\ E_{1y'}(\omega_1)E_{2y'}^*(\omega_2) \end{bmatrix}. \quad (\text{R3})$$

Fig. R2 Schematic of the nonlinear Pancharatnam–Berry (PB) phases for circularly polarized terahertz generations under circularly polarized pump.

Suppose the pump laser field in the global coordinate xyz is circularly polarized with $E_x = 1/2^{1/2}$ and $E_y = \sigma i/2^{1/2}$, where $\sigma = +1$ and $\sigma = -1$ represent LCP and RCP, respectively. Then, the corresponding pump laser field in the local coordinate can be calculated as

$$\begin{bmatrix} E_{x'} \\ E_{y'} \end{bmatrix} = R \begin{bmatrix} E_x \\ E_y \end{bmatrix} = \frac{1}{\sqrt{2}} \begin{bmatrix} \cos\theta & \sin\theta \\ -\sin\theta & \cos\theta \end{bmatrix} \begin{bmatrix} 1 \\ \sigma i \end{bmatrix} = \frac{e^{i\theta}}{\sqrt{2}} \begin{bmatrix} 1 \\ \sigma i \end{bmatrix}, \quad (\text{R4})$$

with R being rotation matrix. Substituting Eq. (R4) into Eq. (R3) gives,

$$\begin{bmatrix} P_{x'}(\omega_{\text{THz}}) \\ P_{y'}(\omega_{\text{THz}}) \end{bmatrix} = \frac{1}{2} \epsilon_0 \begin{bmatrix} \sigma i (\chi_{x'y'x'}^{(2)} - \chi_{x'x'y'}^{(2)}) \\ \chi_{y'x'x'}^{(2)} + \chi_{y'y'y'}^{(2)} \end{bmatrix}. \quad (\text{R5})$$

Transforming Eq. (R5) back to the global coordinate gives,

$$\begin{aligned} \begin{bmatrix} P_x(\omega_{\text{THz}}) \\ P_y(\omega_{\text{THz}}) \end{bmatrix} &= R^{-1} \begin{bmatrix} P_{x'}(\omega_{\text{THz}}) \\ P_{y'}(\omega_{\text{THz}}) \end{bmatrix} = \frac{1}{2} \epsilon_0 \begin{bmatrix} \cos\theta & -\sin\theta \\ \sin\theta & \cos\theta \end{bmatrix} \begin{bmatrix} \sigma i (\chi_{x'y'x'}^{(2)} - \chi_{x'x'y'}^{(2)}) \\ \chi_{y'x'x'}^{(2)} + \chi_{y'y'y'}^{(2)} \end{bmatrix} \\ &= \frac{1}{2} \epsilon_0 \begin{bmatrix} \sigma i (\chi_{x'y'x'}^{(2)} - \chi_{x'x'y'}^{(2)}) \cos\theta - (\chi_{y'x'x'}^{(2)} + \chi_{y'y'y'}^{(2)}) \sin\theta \\ \sigma i (\chi_{x'y'x'}^{(2)} - \chi_{x'x'y'}^{(2)}) \sin\theta + (\chi_{y'x'x'}^{(2)} + \chi_{y'y'y'}^{(2)}) \cos\theta \end{bmatrix}. \end{aligned} \quad (\text{R6})$$

Converting Eq. (R6) into circular polarization basis gives,

$$\begin{aligned} \begin{bmatrix} P_l(\omega_{\text{THz}}) \\ P_r(\omega_{\text{THz}}) \end{bmatrix} &= C \begin{bmatrix} P_x(\omega_{\text{THz}}) \\ P_y(\omega_{\text{THz}}) \end{bmatrix} \\ &= \frac{1}{\sqrt{2}} \begin{bmatrix} 1 & i \\ 1 & -i \end{bmatrix} * \frac{1}{2} \epsilon_0 \begin{bmatrix} \sigma i (\chi_{x'y'x'}^{(2)} - \chi_{x'x'y'}^{(2)}) \cos\theta - (\chi_{y'x'x'}^{(2)} + \chi_{y'y'y'}^{(2)}) \sin\theta \\ \sigma i (\chi_{x'y'x'}^{(2)} - \chi_{x'x'y'}^{(2)}) \sin\theta + (\chi_{y'x'x'}^{(2)} + \chi_{y'y'y'}^{(2)}) \cos\theta \end{bmatrix}, \quad (\text{R7}) \\ &= \frac{1}{2\sqrt{2}} \epsilon_0 \begin{bmatrix} \left[\sigma i (\chi_{x'y'x'}^{(2)} - \chi_{x'x'y'}^{(2)}) + i (\chi_{y'x'x'}^{(2)} + \chi_{y'y'y'}^{(2)}) \right] e^{i\theta} \\ \left[\sigma i (\chi_{x'y'x'}^{(2)} - \chi_{x'x'y'}^{(2)}) - i (\chi_{y'x'x'}^{(2)} + \chi_{y'y'y'}^{(2)}) \right] e^{-i\theta} \end{bmatrix} \end{aligned}$$

where the subscripts l and r represent LCP and RCP, C represents the transforming matrix from linearly polarized basis to circularly polarized basis. **Clearly, nonlinear PB phases of θ for LCP terahertz component and $-\theta$ for RCP terahertz component are observed.**

In general, $\chi^{(2)}_{yyy} = 0$, and $\chi^{(2)}_{yxx}$ is much superior than $\chi^{(2)}_{xyx}$ and $\chi^{(2)}_{xxy}$ in SRR^{R18,R20}, thus Eq. (R7) can be further simplified to,

$$\begin{bmatrix} P_l(\omega_{\text{THz}}) \\ P_r(\omega_{\text{THz}}) \end{bmatrix} = \frac{i\varepsilon_0\chi_{y'x'x'}^{(2)}}{2\sqrt{2}} \begin{bmatrix} e^{i\theta} \\ e^{-i\theta} \end{bmatrix}. \quad (\text{R8})$$

In this case, the LCP and RCP terahertz waves have nearly the same amplitude.

More importantly, the above derivation does not involve any dispersion effect, indicating that the nonlinear PB phase is naturally dispersionless, forming the foundation of the broadband feature of our PTPA.

In the main text, only four SRR orientation angles of 0° , 90° , 180° , and 270° are applied. Therefore, their corresponding nonlinear PB phases of the generated LCP terahertz waves are 0° , 90° , 180° , and 270° , while those of the generated RCP terahertz waves are 0° , -90° , -180° , and -270° , respectively. These are also well validated by our experimental results in Fig. 2g.

We have added the above necessary information in the revised manuscript (see **page 4, paragraph 3; page 5, paragraph 1**) and the revised supplementary information (see **Note 1**).

3. The description of the split-ring resonator (SRR) structure used in this study is also insufficient. Is the SRR structure optimal for terahertz beam deflection control using the nonlinear PB phase? Why were these specific structural parameters chosen? Why is an ITO film present? These questions should be clearly addressed.

Reply: We thank the reviewer for pointing these out. According to our above derivation, nonlinear PB phase is robust. As long as the SRR can generate THz wave, nonlinear PB phase can be obtained by controlling its orientation angle. The specific dimensions of the applied SRR here are $L_1 = 212$ nm, $L_2 = 220$ nm, $L_3 = 110$ nm, $W = 79$ nm, $t_{\text{SRR}} = 40$ nm, period $P_1 = 382$ nm, respectively. These parameters are not specially optimized, but are just chosen based on previous experience and related reports^{R18}, i.e., the strongest terahertz generation happens when the pump wavelength is around the magnetic dipolar resonance (MDR) of the SRR. This is benefitted from the resonance field enhancement effect, which can enhance the efficiency. The above parameters can give rise to a MDR at 1275 nm, where the output power of our optical parametric amplifier (OPA) is around its maximum. We believe that further optimization of the

SRR geometry can potentially improve the terahertz generation efficiency within a certain range. Besides, one can in principle design other geometric parameters to make it work at other pump wavelengths.

As for the ITO film, it has two primary roles:

(1) **Enhance terahertz generation:** Our previous study demonstrates that gold metasurface itself typically exhibit quite low terahertz generation efficiency, incorporating an ultra-thin ITO layer below can strongly boost the efficiency^{R18}. This is arisen from both the MDR field enhancement of the SRR and the epsilon-near-zero (ENZ) field enhancement of the ITO film, manifesting as their coupling effect. In the present case, the ITO film thickness is only 8 nm, which is much thinner than that in Ref. [R18], giving an ENZ wavelength far beyond the MDR. This can be indicated by the measured linear transmission spectrum in Supplementary Fig. S2a, where no distinct coupling induced resonance splitting is observed. Despite this, an enhanced terahertz generation is still observed as compared with the sole SRR case in Ref. [R18], which can be attributed to the larger nonlinear response of the ITO materials and the broadband feature of the ENZ enhancement^{R21}. We believe that the terahertz generation efficiency can be further enhanced by carefully designing the overlap between the ENZ wavelength of ITO and the MDR of the SRR.

(2) **Facilitate fabrication:** Since ITO film is conductive, it can also prevent localized charging effects during electron-beam lithography, which is helpful in increasing fabrication quality, and uniformity, etc.

It should be mentioned that, the main idea of this work is to show the PTPA working scheme, i.e., nonlinear Pancharatnam-Berry (PB) phase and selective excitation concept. This can be done as long as the terahertz generation achieves a detectable level, since the nonlinear PB phase is very robust and solely determined by the orientation angle of the SRR^{R17-R19}.

In summary, there is no denying that better optimization of SRR geometry and ITO thickness can improve the performance by increasing the efficiency and thus the signal-to-noise ratio. However, this does not undermine the underlying physical principles of our approach. We hope our above response can address the concern of the reviewer in this aspect. We have added the above necessary information in the revised manuscript (see **page 5, paragraph 2; page 15, paragraph 1**).

4. On the last paragraph of page 6, the manuscript states that angle-resolved terahertz intensity was measured, but how was this measurement performed? A clear description of the experimental setup and methodology is needed.

Reply: We thank the reviewer for pointing this out. The measurement is performed using the same experimental setup shown in Fig. S3. To measure the angle-resolved terahertz intensity, a moving slit is applied, see the right-bottom part of Fig. S3. As reminded by the reviewer, we notice that the original description is not clear enough, thus we plot the corresponding enlarged view of the core section in Fig. R3.

Fig. R3 Schematic of the core section of the experimental setup in Fig. S3 for measuring the angle-resolved terahertz intensity distribution. PM: parabolic mirror; TLP: THz linear polarizer.

The core section consists of two parabolic mirrors (PM1, $f = 25.4$ mm, and PM2, $f = 50.8$ mm Thorlabs), a 5-mm-wide metallic slit, and two terahertz linear polarizers (TLP1 and TLP2). The PTPA is placed just at the focal point of PM1. When the PTPA performs beamforming functionality, the emitted terahertz waves are directed towards specific angles α determined by the phase gradients, which are subsequently collected and collimated by PM1. It is clear that different emission angle α corresponds to different collimated beam position z after PM1. Nonetheless, after collecting by PM2, they are all focused at the focal point of PM2 for detection. According to the parabolic nature of the reflection surface, one can obtain a relation between the emission angle α and the beam position z ,

$$\alpha = \begin{cases} \tan^{-1}(z \cdot \tan(\beta_1) / 25.4), & z < 0 \\ 0, & z = 0 \\ \tan^{-1}(z \cdot \tan(\beta_2) / 25.4), & z > 0 \end{cases} \quad (\text{R9})$$

Here, $\beta_1 = 36.9^\circ$ and $\beta_2 = 22.6^\circ$ correspond to the collection angle limits on the two sides of the parabolic mirror, respectively. By scanning the position of the slit between PM1 and PM2 along the z direction and recording the transmitted terahertz time-domain signals at each position, the angle-resolved terahertz information can be determined. Since the terahertz beam being controlled here is circularly polarized, two terahertz linear polarizers (TLP1 and TLP2) before PM2 are used to analyze the polarization information at each slit position. During the measurement, TLP2 is fixed to transmit only horizontally polarized terahertz waves, while TLP1 is rotated by either -45° or 45°

to measure orthogonally polarized field information, i.e., E_{45° and E_{-45° . Then, the LCP and RCP terahertz field components E_l and E_r can be extracted using the transforming matrix C in Eq (R7), as well as the corresponding intensities $|E_l|^2$ and $|E_r|^2$.

Based on the above principle, the applied detailed measuring procedures are:

- 1) Set TLP1 to -45° ;
- 2) Scan the slit in a 2-mm step along the z direction, and record the corresponding terahertz time-domain signals, i.e., $E_{-45^\circ}(t, z)$;
- 3) Rotate TLP1 to 45° ;
- 4) Repeat the procedure in step 2) and record $E_{45^\circ}(t, z)$;
- 5) Convert the above measured terahertz time-domain signals to frequency-domain via Fast Fourier Transform (FFT) and extract the field information at the desired frequency f , i.e., $E_{-45^\circ}(f, z)$ and $E_{45^\circ}(f, z)$.
- 6) Calculate the circularly polarized terahertz field components by

$$\begin{bmatrix} E_l(f, z) \\ E_r(f, z) \end{bmatrix} = \frac{1}{\sqrt{2}} \begin{bmatrix} 1 & i \\ 1 & -i \end{bmatrix} \begin{bmatrix} E_{+45^\circ}(f, z) \\ E_{-45^\circ}(f, z) \end{bmatrix}. \quad (\text{R10})$$

- 7) Calculate the final angle-resolved intensity distribution by squaring the field amplitudes ($|E_l|^2$ and $|E_r|^2$) and converting position z to emission angle α using Eq. (R9).

Similar methodology can also be seen in other related works^{R17,R18}. We have added the above necessary information in the revised manuscript (see **page 8, paragraph 2**) and the revised supplementary information (see **Note 4**).

5. Several points regarding the experimental results in Fig. 6 are difficult to understand. Why is the intensity distribution in Fig. 6h not axially symmetric? The manuscript explains that this is due to the discrete nature of the terahertz phase, but even so, the simulation results should still preserve axial symmetry. While it is reasonable that the measured phase distributions in Fig. 6c and Fig. 6g resemble each other, why does the phase distribution in Fig. 6f differ entirely from those in Fig. 6c and Fig. 6g? The same concern applies to the phase distributions in Fig. 6e, Fig. 6h, and Fig. 6i.

Reply: We thank the reviewer for pointing these out. All the simulations here are calculated results using Rayleigh-Sommerfeld diffraction theory by considering each SRR as a point source of the terahertz wave^{R22}. Meanwhile, several aspects in real case are also considered, including the intensity distribution of the pump beam, the distribution of the selectively excited SRRs, and the distance of the calculated results with respect to the PTPA, etc. This will give rise to the non-axial symmetric intensity distributions in Fig. 6h, and the distinct phase distributions in Figs. 6f and 6h. ***The present phase distributions in Figs. 6c and 6e are just equivalent results to intuitively***

show the target functionalities, which are not used in the calculations.

Inspired by these questions, we have carried out more calculations to investigate the underline reasons, which are found to be: **1) the discrete phases induced by selective excitation; 2) the finite size of the PTPA sample.** Figure R4 presents the newly calculated results for the case of Fig. 6h to show the above points. They are corresponding to the generation of LCP terahertz vortex beam with a topological charge of $l = +2$. Without loss of generality, the results are all plotted at 1.0 THz. Since a $4f$ system is applied to measure the response, it is supposed that the measured plane can well rebuild the field distribution around the sample plane. Under such a consideration, the results in Fig. R4 are calculated at 0.7 mm away from the PTPA plane. It should be notice that this would not affect the internal control mechanism.

Fig. R4. Calculated intensity and phase distributions of THz vortex beams (right panel) carrying a topological charge of $l = +2$ at 1.0 THz under various excitation conditions (left panel). The results are obtained using Rayleigh–Sommerfeld diffraction theory at a propagation distance of 0.7 mm.

1) Influence from the discrete phases induced by selective excitation. We start from an ideal case of Gaussian beam incidence with a 0.5-mm spot diameter (denoted as Gauss spot 1) for the pump, and continuous helical phase distribution for the PTPA, see Fig. R4a. Typical donut-shaped vortex intensity and helical phase distributions with good axial symmetry are observed, see Fig. R4b. The intensity distribution can be further homogenized by increasing the sample area. When discretizing the continuous

phases into four levels (0° , 90° , 180° , 270°), see Fig. R4c, the intensity distribution degrades into octagon-shaped while the single phase singularity is divided into two, see Fig. R4d. Even so, the overall distributions still show relatively good axial symmetry.

However, upon further considering the selective excitation pattern to the pump spot by the DMD, see Fig. R4e, the intensity and phase distributions resemble the non-axial symmetric feature in Fig. 6h in our original manuscript, see Fig. R4f. This can be attributed to the uneven sampling of the pump beam in different azimuthal angle ranges, which introduces discrete initial phase distribution.

2) Influence from the finite size of the PTPA sample. The above uneven sampling is more obvious around the PTPA center according to our design. This can be improved by increasing the effective working area. As shown in Fig. R4g, when the working area is expanded to $2 \text{ mm} \times 2 \text{ mm}$ and the Gaussian beam spot is enlarged to 1 mm in diameter (denoted as Gauss spot 2), a larger number of SRRs are excited within the illuminated region. The corresponding intensity and phase distributions show a vortex beam with much better quality in axial symmetry, see Fig. R4h. This can be improved by further increasing the working area through increasing the sizes of the Gaussian spot and the PTPA.

Fig R5. Calculated intensity and phase distributions of THz vortex beams carrying a topological charge of $l = +2$ at 1.0 THz at different propagation distances. **a,b** The intensity and phase distributions at a propagation distance of 2 mm with an imaging area of $4 \text{ mm} \times 4 \text{ mm}$. **c,d** The intensity and phase distributions at propagation distances of 1 mm, 0.7 mm, and 0.5 mm, respectively, with an imaging area of $1.6 \text{ mm} \times 1.6 \text{ mm}$.

Regarding the reviewer's concern on the difference between the calculated phase distributions in Figs. 6f and 6h as compared to those designed and measured results, we find that it is mainly due to the larger propagation distance we set in the previous calculations, which is 2 mm. Considering the small PTPA working area of $1 \text{ mm} \times 1 \text{ mm}$, the emitted terahertz wave (for instance, $\lambda = 300 \mu\text{m}$ at 1.0 THz) has a relatively large divergence angle, which becomes superior for vortex beam. Thus, the

corresponding terahertz waves acquires more propagation phase, especially for the outer fields. This can be indicated by the frequency-dependent phase distributions in Figs. 6f and 6h, where larger frequency acquires larger propagation phase for the same distance.

This discrepancy can be reduced by reducing the propagation distance. Figures R5a and R5b show the originally calculated intensity and phase distributions for case 7 at 1.0 THz in the previous version, where the propagation distance was 2 mm and the imaging area was $4 \text{ mm} \times 4 \text{ mm}$. Figures R5c and R5d illustrate the corresponding results when the propagation distances are 1, 0.7, and 0.5 mm, respectively, where the excitation conditions are kept the same and the imaging areas are all $1.6 \text{ mm} \times 1.6 \text{ mm}$. As the propagation distance increases, the intensity profile enlarges dramatically and keeps a similar shape, while the phase distribution accumulates more spirals. It is found that the results at 0.7 mm are much similar to the measured results in Figs. 6g and 6i, whose imaging areas are also $1.6 \text{ mm} \times 1.6 \text{ mm}$. This also indicates that the measured plane should be closer to the image plane of the fields about 0.7 mm away from the PTPA sample with respect to the $4f$ system.

Fig. R6 Programmable THz vortex beam generation. **a,c** and **b,d** Calculated and measured transverse intensity and phase distributions of the generated LCP THz beams at 0.8, 1.0, 1.2, and 1.4 THz for case 6 and case 7, respectively. The scalebar in the top-left panel of Fig. R6a is applicable to all the results.

Figure R6 replots the corresponding results of Figs. 6f to 6i, where the calculated and measured results are agreeing better with each other. The measured phase inconsistency observed at the outer edges can be attributed to the corresponding weak intensities, where the influence from the noise naturally enhances. Nevertheless, this does not affect the consistency of the main feature of the generated vortex beams.

We thank the reviewer again for these comments, which help us know more feature about this working scheme in 2D phase control. Though the quality of the generated vortex beams still have room for optimization, for example, through increasing the sample size and pump beam spot, it does not affect the underlying mechanism of our working scheme, and our results also clearly demonstrate its effectiveness.

We have added the above necessary information in the revised manuscript (see **page 13, paragraph 2; page 14 paragraph 1; page 24, Fig. 6**) and the revised supplementary information (see **Note 7 and Fig. S10**).

Reference

- [R1-1] Yang, Y., Gurbuz, O. D. & Rebeiz, G. M. An Eight-Element 370–410-GHz Phased-Array Transmitter in 45-nm CMOS SOI With Peak EIRP of 8–8.5 dBm. *IEEE Trans. Microwave Theory Tech.* **64**, 4241-4249 (2016).
- [R1-2] Monnai, Y., Lu, X. & Sengupta, K. Terahertz Beam Steering: from Fundamentals to Applications. *J. Infrared, Millimeter, Terahertz Waves* **44**, 169-211 (2023).
- [R1-3] Fu, X., Yang, F., Liu, C., Wu, X. & Cui, T. Terahertz Beam Steering Technologies: From Phased Arrays to Field-Programmable Metasurfaces. *Adv. Opt. Mater.* **8**, 1900628 (2019).
- [R1-4] Boyd, R. W. *Nonlinear Optics* (Academic Press, 2020).
- [R1-5] Kamiura, Y. et al. 300-GHz Beam-Steering Wireless Communication Enabled by 4-Array InGaAs UTC-PD on SiC Substrate and Optical Phased Array. *J. Lightwave Technol.* , 1-8 (2025).
- [R1-6] Froberg, N., Bin, B., Zhang, X. & Auston D.H. Terahertz radiation from a photoconducting antenna array. *IEEE J. of Quantum Electronics* **28**, 2291-2301 (1992).
- [R1-7] Che, M., Kondo, K., Kanaya, H.&Kato, K. Arrayed Photomixers for THz Beam-Combining and Beam-Steering. *J. Lightwave Technol.* **40**, 6657-6665 (2022).
- [R1-8] Sharma, M., Tal, M., McDonnell, C. & Ellenbogen, T. Electrically and all-optically switchable nonlocal nonlinear metasurfaces. *Sci. Adv.* **9**, eadh2353 (2023).
- [R1-9] Hu, Y., et al. Electrically Tunable Multifunctional Polarization-Dependent Metasurfaces Integrated with Liquid Crystals in the Visible Region. *Nano Lett.* **21**, 4554-4562 (2021).
- [R1-10] Benford, J., Swegle, J. A. & Schamiloğlu, E. *High power microwaves* (Taylor and Francis, New York, 2007).
- [R1-11] Wang, S. et al. Flexible generation of structured terahertz fields via programmable exchange-biased spintronic emitters. *eLight* **4**, 11 (2024).

- [R1-12] Berry, C. W., Wang, N., Hashemi, M. R., Unlu, M. & Jarrahi, M. Significant performance enhancement in photoconductive terahertz optoelectronics by incorporating plasmonic contact electrodes. *Nat. Commun.* **4**, 1622 (2013).
- [R1-13] Chen, C., Chen, S., Ni, Y., Xu, Y. & Yang, Y. Liquid Crystal Metasurface for On-Demand Terahertz Beam Forming Over 110° Field-Of-View. *Laser Photonics Rev.* **18**, 2400237 (2024).
- [R1-14] Chen, B. et al. Electrically addressable integrated intelligent terahertz metasurface. *Sci. Adv.* **8**, eadd1296 (2022).
- [R1-15] Zeng, H. et al. Ultrafast modulable 2DEG Huygens metasurface. *Photonics Res.* **12**, 1004-1015 (2024).
- [R1-16] Li, M. et al. Lithium niobate photonic-crystal electro-optic modulator. *Nat. Commun.* **11**, 4123 (2020).
- [R1-17] McDonnell, C., Deng, J., Sideris, S., Ellenbogen, T. & Li, G. Functional THz emitters based on Pancharatnam-Berry phase nonlinear metasurfaces. *Nat. Commun.* **12**, 30 (2021).
- [R1-18] Lu, Y. et al. Integrated Terahertz Generator-Manipulators Using Epsilon-near-Zero-Hybrid Nonlinear Metasurfaces. *Nano Lett.* **21**, 7699-7707 (2021).
- [R1-19] Wang, Q. et al. Nonlinear Terahertz Generation: Chiral and Achiral Meta-Atom Coupling. *Adv. Funct. Mater.* **33**, 2300639 (2023).
- [R1-20] Jia, W. et al. Polarization-entangled Bell state generation from an epsilon-near-zero metasurface. *Sci. Adv.* **11**, eads3576 (2025)
- [R1-21] Jia, W. et al. Broadband terahertz wave generation from an epsilon-near-zero material. *Light Sci. Appl.* **10**, 11 (2021).
- [R1-22] Wang, Q. et al. Broadband metasurface holograms: toward complete phase and amplitude engineering. *Sci. Rep.* **6**, 32867 (2016).

Reviewer #2:

The authors present a novel technique for terahertz beamforming and steering using a massive array of nonlinear meta-elements with varying orientations, selectively excited by a circularly polarized optical pump. While the reviewer is impressed by the technical advancements, the authors should consider the following points.

Reply: We sincerely thank the reviewer for the positive evaluation of our work, and we are honored that our technical advancements can make a favorable impression. We hope our following responses can address the reviewer's comments and concerns.

1. The title of the paper "Photonic Terahertz Phased Array" appears to be an overstatement and does not accurately reflect the essence of the technical contribution by the authors. Indeed, there have been papers on photonic terahertz phased arrays not cited in the current manuscript such as

-Froberg, Nan Moore, et al. "Terahertz radiation from a photoconducting antenna array." *IEEE Journal of Quantum Electronics* 28.10 (1992): 2291-2301.

-Maki, Ken-ichiro, and Chiko Otani. "Terahertz beam steering and frequency tuning by using the spatial dispersion of ultrafast laser pulses." *Optics Express* 16.14 (2008): 10158-10169.

-Che, Ming, et al. "Arrayed photomixers for THz beam-combining and beam-steering." *Journal of Lightwave Technology* 40.20 (2022): 6657-6665.

I admit that the authors have done a great work to utilize Pancharatnam-Berry (PB) phase, which enables to make the output phase controllable with the orientation of SRRs, while the previous photonic terahertz phased arrays had to rely on the true time delay or the phase shift of the optical pump. The authors should emphasize their technical advancement in comparison to those previous works.

Reply: We would like to once again thank the reviewer for the positive evaluation of our approach, especially for the kind remark that we have done "a great work to utilize PB phase." We apologize that the previous version of our manuscript did not sufficiently review the existing studies on photonic terahertz phased arrays (PTPAs). We are also grateful to the reviewer for providing these classic references, which represent three distinct strategies for controlling the direction of the radiated terahertz waves. We have carefully read and studied these works, and the corresponding beam control mechanisms are summarized as follows:

1) Photoconductive Antenna Array Method: The work by Froberg et al. (1992) demonstrates a photoconductive antenna array capable of generating electrically steerable terahertz radiation. The array is pumped by a femtosecond pulse train with fixed time delay to narrow the terahertz bandwidth, and the terahertz beam direction is steered by changing the period of the spatially sinusoidal bias voltages to the associated electrode array.

2) Spatial Dispersion Control Method: The work by Maki et al. (2008) demonstrates a terahertz beam steering and frequency tuning method using simple stripe-line photoconductive antenna pumped by two beams of spatially dispersed ultrafast laser pulses on the basis of difference frequency generation process. The terahertz beam direction is steered via tilting one pump beam, whereas the frequency is tuned by laterally shifting the two overlapping beams. The method eliminates phase shifters and array structures.

3) Photomixer Array Method: The work by Che et al. (2022) presents a monolithic InP/InGaAs-based 1×4 photomixer array. Each photomixer contains a UTC-PD with integrated 1×4 planar slot antennas. Both continuous and pulsed terahertz beams can be generated depending on the types of the pump lasers. By increasing the number of the excited UTC-PDs, the terahertz radiation power can be enhanced through beam-combing effect. By controlling the gradient delays of the pump lasers to the 4 UTC-PDs, the terahertz beam direction can be steered.

In comparison, the photoconductive antenna array method is essentially an amplitude-based modulation approach, which involves at most 2-level phase control (0 and 180°) induced by the direction of the bias voltage. As a result, it often generates symmetric terahertz diffraction beams. While this method allows for convenient scaling of the array size, the number of required electrodes increases accordingly, introducing significant challenges in fabrication and system control.

The spatial dispersion control method is a clever utilization of the difference-frequency generation process. By designing spatial dispersion into the optical setup, it eliminates the need for antenna arrays or additional phase control elements. However, the optical configuration becomes relatively complex. Meanwhile, the terahertz beam steering angle and frequency can be very sensitive to variations in the pump beams, which increases the requirements on the optical stability. Moreover, it faces potential challenges in generating and controlling multiple THz beams simultaneously.

As for the photomixer array method, element based on UTC-PD is currently one of the most promising devices for achieving terahertz communications. However, the fabrication cost is typically high, especially when integrating multiple UTC-PDs into a dense array. Although integrating each UTC-PD with multiple antennas can reduce the required number of UTC-PDs, the radiation bandwidth inherent to the antenna design limits the available terahertz bandwidth. Furthermore, the approach depends on external

optical delay to control the terahertz phase, which becomes increasingly complex and costly as the number of array elements grows.

In summary, as the reviewer pointed out, all the above methods rely on true optical delay or phase shift of the optical pump to achieve terahertz phase control. Additionally, these approaches are constrained by limited degrees of freedom and array scalability, and have so far only demonstrated single-beam control and dual-beam control with a fixed angle relation. They are also currently limited to one-dimensional (1D) control, and requiring further investigations to overcome the difficulty in implementing two-dimensional (2D) control. *In contrast*, our proposed approach is based on nonlinear PB phase control and selective excitation concept. It requires neither actual time delay nor phase shifts of the optical pump, and supports multi-level phase control, large-scale integration, and 2D programmable beam forming. As such, our approach overcomes the difficulties of the aforementioned methods, and offers a versatile and scalable solution for advanced terahertz wavefront engineering.

Regarding the title of this work, we agree the reviewer's opinion that it does not accurately reflect the essence of the technical contribution. We have revised it to "*Photonic terahertz phased array via selective excitation of nonlinear Pancharatnam-Berry elements*".

We thank the reviewer for providing these representative references. We have added the above necessary discussion and comparison to emphasize our technical advancements and cited them as Refs. 15-17 in the revised manuscript (see **page 1, title; page 3, paragraph 2; page 19, Fig. 1**).

2. The implementation of the present phased array is based on the micromachine technology, i.e. DMD for the light control. In that case, the authors should also discuss the advantages and disadvantages of their approach in comparison with other terahertz phased arrays based on micromachines such as

- Busch, Stefan, et al. "Optically controlled terahertz beam steering and imaging." *Optics letters* 37.8 (2012): 1391-1393.

- Monnai, Yasuaki, et al. "Terahertz beam steering and variable focusing using programmable diffraction gratings." *Optics express* 21.2 (2013): 2347-2354.

- Liu, Xuan, et al. "Terahertz beam steering using a MEMS-based reflectarray configured by a genetic algorithm." *IEEE Access* 10 (2022): 84458-84472.

Reply: We thank the reviewer for pointing this out and providing these relevant

references. As the reviewer mentioned, these works and ours both rely on micromachine technology for achieving terahertz beam control.

1) The work by Busch et al. (2012) demonstrates an optically controlled broadband spatial terahertz modulator using light-induced free carriers in silicon. The basic idea is using a digital white light projector to create 2D plasma patterns in silicon through free carrier excitation, which can modulate the spatial terahertz transmission via carrier absorption. Terahertz beam steering is achieved through inducing virtual grating structures with varying grating periods.

2) The work by Monnai et al. (2013) presents a programmable terahertz diffraction grating using electrostatically actuated metallic cantilever array. By reconfiguring the array with periodic and chirped grating patterns, terahertz beam steering and variable focusing are achieved via diffraction effect, respectively.

3) The work by Liu et al. (2022) presents a genetic algorithm that is adapted to MEMS-based THz reflectarray. By optimizing the height profile of 80 individually actuated reflector elements using the algorithm, continuous beam steering, sidelobe suppression, and null control are achieved, which are otherwise challenging for traditional grating-based reflectarrays. The method allows compact and efficient terahertz beamforming without complex feeding networks.

In summary, these works are excellent examples of micromachine-enabled dynamic terahertz beam control. However, their working mechanisms lie in those of the linear metasurfaces that we mentioned in the main text, which rely on the linear interaction between the incident terahertz waves and the devices, i.e., working in a terahertz-in/terahertz-out mode. In specific, the first two works rely on transmission amplitude control, while the third one relies on reflection phase control. Compared with linear metasurfaces, these micromachine-based approaches typically operate with non-resonant structures and therefore offer broader bandwidth. However, they also suffer from inherent insertion losses due to the interaction with the terahertz field. **By contrast**, our work follows a pump-in/terahertz-out mode, which leverages the nonlinear interaction between the optical pump and the device, thereby enabling simultaneous terahertz generation and control. This integrated scheme naturally supports broadband operation while bypasses the issue of terahertz insertion loss.

Admittedly, our approach involves nanofabrication, which currently limits the sample size compared to the above approaches and results in larger beam divergence and reduced angular resolution. However, these limitations can be mitigated through large-area fabrication techniques, such as nanoimprint lithography and pattern stitching technology. On the other hand, the above approaches can in principle work with powerful external terahertz sources for specific high-power applications, where the final terahertz output intensity may be higher than ours without considering the large insertion loss. It should be mentioned that, our strategy is also adaptable to more

efficient terahertz generation platforms, such as spintronic emitters^{R1}, photoconductive antenna^{R2}, and photomixers^{R3}, etc., to get larger terahertz output intensity.

We thank the reviewer again for providing these relevant references. We have also added the above necessary discussion and comparison to emphasize the advantages of our approach and cited them as Refs. 25-27 in the revised manuscript (see **page 3, paragraph 3; page 15, paragraphs 1 and 3; page 16, paragraph 1**).

3. The authors calculate the array factor in Eq. (2) to explain the far-field pattern in specific cases. However, characterizing a phased array requires discussing both the angular resolution and the steerable range of the beam, which must be dependent on the geometry of the SRR array. This analysis is crucial for understanding the necessary improvements to enhance performance in the future.

Reply: We thank the reviewer for pointing this out. Yes, both the angular resolution and the steerable range are key parameters in characterizing a phased array.

1) Angular Resolution: The angular resolution reflects the system's ability to distinguish between two closely spaced angular directions, which is primarily determined by the far-field beamwidth. Larger resolution corresponds to smaller beam width. For our 1D demonstration, the beamwidth depends on the effective aperture length of the array, which is given by $L = N_x \cdot P_2$, where N_x is the number of sub-elements (SRR array) along the x -direction, and $P_2 = 50 \mu\text{m}$ is the period of sub-element. The theoretical far-field beamwidth can be expressed as^{R4}:

$$\Omega_{theo.} = \frac{\lambda}{L \cdot \cos(\alpha)} = \frac{\lambda}{N_x \cdot P_2 \cdot \cos(\alpha)}. \quad (\text{R11})$$

where λ is the central wavelength and α is the beam deflection angle. To more intuitively illustrate the influence of array length on beamwidth, we perform beamwidth calculations for arrays with aperture lengths L of 2 mm, 4 mm, and 8 mm (corresponding to $N_x = 40, 80, 160$) at $\lambda = 300 \mu\text{m}$ (1.0 THz) as an example, see Fig. R7a. It is seen that the beamwidth increases as the absolute deflection angle increases, while decreases as the aperture length increases.

In this work, the sample has a total length of 2 mm, corresponds to the black curve in Fig. R7a. The green squares and yellow triangles represent case 1 and case 2 in the main text, respectively. The corresponding theoretical beamwidths are $\Omega_{theo.} = 8.7^\circ$ and 9.3° . To more clearly compare how the aperture length affects beamwidth, we further calculated the far-field radiation patterns using Eq. (2), as shown in Figs. R7b and R7c. The resulting beamwidths Ω_{cal} for case 1 and case 2 are observed to decrease from 7.8° and 8.3° (at $L = 2 \text{ mm}$) to 3.9° and 4.1° (at $L = 4 \text{ mm}$), and finally to 1.9° and 2.0° (at L

= 8 mm). These results, represented by the red squares and blue triangles, are consistent with the theoretical prediction, see Fig. R7a. It should be noticed that the beam steering angle here cannot be continuous or tuned using the theoretical angular resolution since discrete phase control is applied.

Fig. R7. Calculated beamwidths for PTPAs with different aperture lengths of 2 mm, 4 mm, and 8 mm. **a** Theoretical beamwidth at various emission angles. **b,c** Calculated far-field radiation patterns using the phase control schemes of case 1 and case 2 in the main text at 1.0 THz.

2) Steerable range: The steerable range refers to the maximum angular span over which a phased array can effectively direct its main radiation beam while maintaining beam quality. This range is fundamentally constrained by the sub-element spacing and the phase gradient across the PTPA here. In our case, the sub-element spacing is $P_2 = 50 \mu\text{m}$, which satisfies $P_2 < \lambda/2$ for the central wavelength $\lambda = 300 \mu\text{m}$. This criterion effectively suppresses the formation of grating lobes within the designed steering range.

Considering the case of single beam steering, the maximum steerable range of a beam with wavelength λ is determined by the achievable Λ , according to the generalized Snell's law, i.e., $\alpha = \arcsin(\lambda/\Lambda)^{R5}$. The beam steering here is implemented using four discrete sub-element phase states. To obtain a relative uniform linear phase gradient for efficient beam steering, the span of each phase state should better be the same, this constraint results in a phase period $\Lambda = 4mP_2$ with $m = 1, 2, 3, \dots$

For a wavelength of $\lambda = 300 \mu\text{m}$, the minimum $\Lambda_{\min} = 8P_2 = 400 \mu\text{m}$, corresponding to a maximum steerable range of $-48.6^\circ \sim 48.6^\circ$. Figure R8a illustrates the calculated far-field radiation patterns of different Λ using Eq. (2), where the angular-dependent gain is taken into account. It should be mentioned that larger steerable range is in principle achievable, if we ignore all the above constraints and apply non-uniform linear phase distribution, see Fig. R8b. The steerable range is enlarged to $-59^\circ \sim 59^\circ$. The corresponding DMD coding scheme for different phase states are schematically illustrated in Fig. R8c. The sub-element size P_2 is $50 \mu\text{m}$. By selectively exciting the nonlinear PB sub-elements of desired orientations (denoted by different colors), terahertz waves with the corresponding phase can be generated. Based on this coding

scheme, different phase periods Λ with uniform and non-uniform phase profiles are designed, as shown in Figs. R8d and R8e, respectively. Taking LCP THz waves as an example, such phase distributions enable beam steering toward the negative direction. By simply mirroring these spatial arrangements along the y direction, beam steering toward the positive direction can be achieved. When the phase distribution remains constant, as shown in Fig. R8f, the beam radiates towards the normal direction.

Fig. R8 Calculated far-field radiation patterns of the PTPA using different phase periods Λ with uniform **a** and non-uniform **b** phase distributions, respectively. **c** Schematic of the DMD coding scheme for different phase states, where the solid and hollow squares represent excited and non-excited nonlinear PB sub-elements, respectively. **d,e** Equivalent uniform and nonuniform phase distributions of different phase periods Λ for calculating corresponding results of $k_x < 0$ in **a** and **b**, respectively. **f** Equivalent constant phase distribution of $k_x = 0$.

Notice that, the beam steering angle cannot be continuously tuned, owing to the discrete feature of the phase control scheme. However, the angle step can be potentially reduced using smaller sub-element size P_2 , which can be simply achieved here since the SRR size is only $P_1 = 382$ nm. This can also increase the steerable range based on the above control manner of the phase period, as smaller Λ can be achieved.

In summary, the angular resolution is determined by the aperture length L of the PTPA, which can be improved using larger L , whereas the beam steerable range is determined by the achievable phase period Λ , which can be improved using smaller sub-element size P_2 . They are clearly not contradicting in our design. Other optimization methods may include applying certain designing algorithms of the phase distributions^{R6}. Besides, our approach also in principle supports terahertz amplitude control through controlling the excitation area within each sub-element, which could serve as another degree of freedom in improving the performance. Though the above analyses are carried out by taking 1.0 THz as an example, the same conclusion can be extended to the other terahertz frequencies.

We have added the above necessary information in the revised manuscript (see **page 8, paragraph 3; page 9, paragraph 1**) and the revised supplementary information (see **Note 5 and Note 6**).

4. What is the power conversion efficiency of the proposed nonlinear metasurface? The authors should also discuss any potential trade-offs between the beam steering performance and efficiency.

Reply: We thank the reviewer for pointing this out. Since we are currently lacking the condition in directly measuring the power of the generated terahertz wave, we choose to evaluate the efficiency level indirectly through comparing the terahertz generation of the nonlinear metasurface with that of a traditional ZnTe crystal. Without loss of generality, the nonlinear metasurface sample is a uniform SRR array with the same orientation, while the ZnTe crystal is <110>-cut with a thickness of 200 μm .

Figure R9a shows the corresponding measured terahertz peak-to-peak amplitudes as a function of pump wavelength at 1275 nm, where the result of the ZnTe crystal under 800 nm pump is also presented. It is seen that the responses of the ZnTe crystal under 1275 nm and 800 nm pumps are nearly the same, while the response of the nonlinear metasurface exhibits a comparable level, although a saturation effect emerges at higher fluences. This observation aligns with previously reported results^{R7-R9}.

Fig. R9 Measured pump-fluence-dependent THz peak-to-peak amplitudes of the nonlinear metasurface sample under 1275 nm pump, and a 200 μm -thick ZnTe crystal under 800 nm and 1275 nm pump. **b** Measured time-domain signals of the sample and ZnTe crystal (at 800 nm) at $56.6 \mu\text{J cm}^{-2}$ pump fluence.

According to Ref. [R10], a 500 μm -thick ZnTe crystal achieves a conversion efficiency of 3×10^{-5} under a 1.4 mJ/cm^2 pump at 800 nm. In our case, the terahertz peak-to-peak amplitude of the nonlinear metasurface is approximately 0.58 of that of the ZnTe under a $56.6 \mu\text{J/cm}^2$ pump, as shown in Fig. R9b. Taking into account the differences in pump fluence, sample thickness, and relative terahertz field amplitude, the efficiency is roughly estimated to be $3 \times 10^{-5}/(25 \times 2.5^2 \times 1.72^2) = 6.5 \times 10^{-8}$, where the factors of 25, 2.5, and 1.72 correspond to the pump fluence ratio, thickness ratio of ZnTe in Ref.[R10] and here, and terahertz field amplitude ratio of ZnTe and nonlinear metasurface in Fig. R9b, respectively. Although the absolute efficiency is modest, it is critical to emphasize that our nonlinear metasurface is only 48 nm thick, which is four-orders thinner than the ZnTe crystal. This implies a remarkably high effective nonlinear response. In specific, the effective second-order susceptibility of the nonlinear metasurface can be ~ 541 times of ZnTe.

It is clear that higher efficiency gives rise to stronger terahertz radiation, which in turn improves the dynamic range and signal-to-noise ratio of the PTPA, and thus the beam steering performance. **Regarding the potential trade-offs between performance and efficiency of our nonlinear metasurface:** the first one should be the saturation effect, which means that the efficiency cannot keep a large increasing trend as the pump fluence; the second one should be the low damaging threshold feature of nonlinear metasurfaces based on plasmonic structures, where excessive pump fluence will bring irreparable damage to the sample.

We have added the above necessary information in the revised manuscript (see page 14, paragraph 3; page 15, paragraph 1) and the revised supplementary information (see Note 8).

5. The proposed principle may inherently require a short-pulse optical pump, but is there any potential for applying this technique to a CW optical pump? Note that the previous works mentioned above are generally compatible with both pulsed and CW optical pumps.

Reply: We thank the reviewer for pointing this out. In terms of the current nonlinear metasurface approach, it is challenging to work with a CW optical pump. For example, the femtosecond laser pulse employed in our experiments has a peak power of approximately 16.7 MW, which is difficult to achieve with a CW laser. It is certain that some terahertz radiation can still be generated under a common watt-level CW pump, but the signal would be extremely weak and hardly detectable.

However, it is worth mentioning that although the nonlinear metasurface approach is challenging, our scheme, i.e., nonlinear PB phase control + selective excitation concept, is applicable, for example, in photoconductive-antenna-based photomixers, which can work with a CW pump^{R11}. As nonlinear PB phase is a robust phase control method, the radiation phases of LCP and RCP terahertz waves are solely determined by the orientation angle of the emission structure. This is also true for photomixer junction with different orientation angles (fixed bias direction). Therefore, by arranging these junctions to form an array with a similar strategy to ours, the proposed PTPA functionalities here can also be realized by employing similarly selective optical excitation method, or even selective electric-bias excitation method. Of course, a key challenge of such a device lies in the design of electrode configurations. Nevertheless, recent advances in bias-free two-metal Schottky junctions may provide a promising pathway to circumvent the need for external voltage sources^{R12}.

We have added the above necessary discussion in the revised manuscript (**see page 15, paragraph 2**).

6. While the authors claim the importance of beam steering in various terahertz applications stating in Line48 “a wide range of advanced technological systems and applications ranging from radar, communication, and astronomy. [3,4],” the cited literature seems to be only about communication, not covering radar and astronomy. The authors might want to add literature on them.

Reply: We thank the reviewer for pointing this out. It is our oversight as we paid more attention on communication-related applications. To provide a more balanced overview, we have now added the following two references on terahertz beam steering applications in radar and astronomy:

1) Matsumoto, H., Watanabe, I., Kasamatsu, A. *et al.* Integrated terahertz radar based on leaky-wave coherence tomography. *Nat. Electron.* **3**, 122–129 (2020).

This paper demonstrates an integrated terahertz radar system based on leaky-wave coherence tomography, which enables beam steering and homodyne detection without

using phase shifters, circulators, or mechanical scanners. By employing a pair of reverse-connected leaky-wave antennas operating in the 330–500 GHz range, the system can detect both the direction and distance of a target through frequency-swept measurements. The authors showcase high angular resolution, millimeter-level ranging capability, and even remote human heartbeat detection through clothing.

2) F. Teng, J. X. Wan, and J. Liu, “Review of terahertz antenna technology for science missions in space,” **IEEE Aerosp. Electron. Syst. Mag.** **38(2)**, 16–32 (2023).

This review highlights the important role of terahertz antenna technology in space-based astronomy space missions. It discusses several key examples, including the Submillimeter Wave Astronomy Satellite for observing molecular lines in interstellar clouds, the Herschel Space Observatory for far-infrared and submillimeter astronomy, the Planck satellite for cosmic microwave background studies, and the Microwave Instrument for the Rosetta Orbiter instrument for comet analysis. These missions demonstrate how terahertz antennas enable high-resolution and high-sensitivity observations that are essential for studying the structure and evolution of the universe.

We have cited them as Refs. 5 and 6 in the revised manuscript (see page 2, paragraph 2).

7. The following sentence in Line123 seem to be incomplete.

“Although the excited sub-elements are positioned differently, which induces phase differences for emitted THz waves towards oblique directions.”

Reply: We thank the reviewer for pointing this out. This sentence is meaningful only when paired with the sentence behind it, i.e., “Given that the sub-elements are subwavelength and their spatial offsets are small compared to the element size, this phase crosstalk is not considered in Fig. 2d for simplicity.”

In our design, the four types of sub-elements are all not uniformly arranged within each element, this would cause additional non-uniform propagation phase differences for oblique terahertz radiation, which are also known as the “detour phase”^{R13}. For an arbitrary 2D deflection angle (θ_x, θ_y) , the detour phase can be calculated as $\varphi_d = k(x\sin\theta_x + y\sin\theta_y)$ with x and y being the location of the sub-element.

1) For the case of normal and small-angle terahertz radiation, the detour phase is small. The phase of each sub-element is sole determined by the nonlinear PB phase. Thus, when the sub-elements of the same orientation being excited, their nonlinear PB phase

can roughly be considered as the phase of each element, see Fig. 2d in the main text.

2) For the case of beam steering along the x or the y directions, where $\theta_y = 0$ or $\theta_x = 0$, the terahertz radiations from the sub-elements of the same orientation in each element can be expressed as $\exp(i\varphi_{\text{PB}})[1 + \exp(i\varphi_d) + \exp(i2\varphi_d) + \exp(i3\varphi_d)]$ with $\varphi_d = kP_2\sin\theta_x$ or $kP_2\sin\theta_y$, respectively, according to the arrangement style of the sub-elements in Fig. 2 (the detour phases are referenced to the one at the element edge). It is seen that the detour phase contributions are the same, the nonlinear PB phase can still be considered as the phase of each element.

3) Whereas for the case of the other deflection angles, the detour phases will affect, whose contribution is determined by the actual angle and will make the final radiation phase fluctuates around the nonlinear PB phase.

In Fig. 2, only nonlinear PB phase is considered to show our phase control scheme in a clear physical picture, since it is the main source of the phase control here. However, to mention the existence of the detour phase, we add such a sentence in the main text. Notice that, the detour phase contribution is considered in all the theoretical calculations in this work to give a more rigorous PTPA performance.

We noticed that the original sentence did not clearly convey our intended meaning, we have revised the sentence as follows:

“Although the excited sub-elements are positioned differently within the element, potentially introducing detour phase differences for emitted THz waves towards oblique directions, the main phase contribution is still from the nonlinear PB phase (see Supplementary Note 2). This phase crosstalk is not considered in Fig. 2d for simplicity, which would not affect the fundamental controlling picture.”

We have added the above necessary information in the revised manuscript (see page 6, paragraph 1) and the revised supplementary information (see Note 2).

Reference

- [R2-1] Wang, S. et al. Flexible generation of structured terahertz fields via programmable exchange-biased spintronic emitters. *eLight* **4**, 11 (2024).
- [R2-2] Berry, C.W., Wang, N., Hashemi, M.R., Unlu, M. & Jarrahi, M. Significant performance enhancement in photoconductive terahertz optoelectronics by incorporating plasmonic contact electrodes. *Nat. Commun.* **4**, 1622 (2013).
- [R2-3] Che, M., Kondo, K., Kanaya, H. & Kato, K. Arrayed Photomixers for THz Beam-Combining and Beam-Steering. *J. Lightwave Technol.* **40**, 6657-6665 (2022).
- [R2-4] Mailloux, R. *Phased Array Antenna Handbook* (Artech House, 2017).

- [R2-5] Yu, N. et al. Light Propagation with Phase Discontinuities: Generalized Laws of Reflection and Refraction. *Science* **334**, 333-337 (2011).
- [R2-6] Wen, Y., Wang, B. & Ding, X. A Wide-Angle Scanning and Low Sidelobe Level Microstrip Phased Array Based on Genetic Algorithm Optimization. *IEEE Trans. Antennas Propag.* **64**, 805-810 (2016).
- [R2-7] McDonnell, C., Deng, J., Sideris, S., Ellenbogen, T. & Li, G. Functional THz emitters based on Pancharatnam-Berry phase nonlinear metasurfaces. *Nat. Commun.* **12**, 30 (2021).
- [R2-8] Lu, Y. et al. Integrated Terahertz Generator-Manipulators Using Epsilon-near-Zero-Hybrid Nonlinear Metasurfaces. *Nano Lett.* **21**, 7699-7707 (2021).
- [R2-9] Wang, Q. et al. Nonlinear Terahertz Generation: Chiral and Achiral Meta-Atom Coupling. *Adv. Funct. Mater.* **33**, 2300639 (2023).
- [R2-10] Blanchard, F. et al. Generation of 1.5 μ J single-cycle terahertz pulses by optical rectification from a large aperture ZnTe crystal. *Opt. Express* **15**, 13212-13220 (2007).
- [R2-11] Tanoto, H. et al. Nano-antenna in a photoconductive photomixer for highly efficient continuous wave terahertz emission. *Sci. Rep.* **3**, 2824 (2013).
- [R2-12] McBryde, D. et al. Multiple double-metal bias-free terahertz emitters. *Appl. Phys. Lett.* **104**, 201108 (2014).
- [R2-13] Deng, Z. et al. Full-Color Complex-Amplitude Vectorial Holograms Based on Multi-Freedom Metasurfaces. *Adv. Funct. Mater.* **30**, 1910610 (2020).

Response to the Reviewers

We sincerely thank both reviewers for their time, thoughtful feedback, and constructive suggestions throughout the review process. Their insightful comments have greatly contributed to the clarity, quality, and overall presentation of our work.

Reviewer #1:

The manuscript has been appropriately revised in accordance with my comments. The novelty of the work as a terahertz light source is now clearly conveyed, and the technical concerns have been resolved. I appreciate the authors' sincere response, and I recommend this manuscript for publication in Nature Communications.

Reply: We are very happy that our revisions can satisfy the high criteria of the reviewer, and are grateful to the reviewer for recommending its publication in Nature Communications.

Reviewer #2:

The authors have revised their manuscript, fully addressing the review comments. Hence, I recommend its acceptance.

Reply: We are very happy that our revisions have addressed the comments of the reviewer, and are grateful to the reviewer for recommending its acceptance.